# TRIP-Bench: A Benchmark for Long-Horizon Interactive Agents in Real-World Scenarios

**Yuanzhe Shen** [* 1 2] **Zisu Huang** [* 2] **Zhengyuan Wang** [* 2] **Muzhao Tian** [* 2] **Zhengkang Guo** [2] **Chenyang Zhang** [3] **Shuaiyu Zhou** [4] **Zengjie Hu** [4] **Dailin Li** [5] **Kaimin Wang** [2] **Wenhao Liu** [6] **Tianlong Li** [1] **Feng Hong** [1] **Cao Liu** [1] **Ke Zeng** [1]

## Abstract

As LLM-based agents are deployed in increasingly complex real-world settings, existing benchmarks underrepresent key challenges such as enforcing global constraints, coordinating multi-tool reasoning, and adapting to evolving user behavior over long, multi-turn interactions. To bridge this gap, we introduce **TRIP-Bench**, a long-horizon benchmark grounded in realistic travel-planning scenarios. TRIP-Bench leverages real-world data, offers 18 curated tools and 40+ travel requirements, and supports automated evaluation. It includes splits of varying difficulty; the hard split emphasizes long and ambiguous interactions, style shifts, feasibility changes, and iterative version revision. Dialogues span up to 15 user turns, can involve 150+ tool calls, and may exceed 200k tokens of context. Experiments show that even advanced models achieve at most 50% success on the easy split, with performance dropping below 10% on hard subsets. We further propose **GTPO**, an online multi-turn reinforcement learning method with specialized reward normalization and reward differencing. Applied to Qwen2.5-32B-Instruct, GTPO improves constraint satisfaction and interaction robustness, outperforming Gemini-3-Pro in our evaluation. We expect TRIP-Bench to advance practical long-horizon interactive agents, and GTPO to provide an effective online RL recipe for robust long-horizon training.

[*]Equal contribution [1]LongCat Interaction Team, Meituan, Shanghai, China [2]School of Computer Science, Fudan University, Shanghai, China [3]Wuhan University, Wuhan, China [4]Peking University, Beijing, China [5]Dalian University of Technology, Dalian, China [6]Xiaohongshu Inc., Shanghai, China. Correspondence to: Feng Hong <hongfeng03@meituan.com>, Yuanzhe Shen <yzshen25@m.fudan.edu.cn>.

*Proceedings of the 43rd International Conference on Machine Learning*, Seoul, South Korea. PMLR 306, 2026. Copyright 2026 by the author(s).

## 1. Introduction

In recent years, Large Language Models (LLMs) have advanced in reasoning, planning, and tool use (DeepSeek-AI et al., 2025; Bai et al., 2025; Zeng et al., 2025), accelerating the deployment of LLM-based agents in real applications (Hager et al., 2024; Cheng et al., 2025). As agents shift from "answering questions" to "completing tasks," they must produce executable and revisable action sequences and sustain progress toward long-horizon goals—raising requirements for reasoning depth, planning quality, and cross-turn decision consistency. Real deployments further impose predefined rules, workflow and compliance constraints (Qi et al., 2025), while user instructions and preferences evolve through interaction and are rarely fully specified upfront. Consequently, agents must align local decisions with global constraints and remain consistent and controllable throughout multi-turn, dynamic processes. These realities make multi-turn task completion a central dimension of agent evaluation, motivating benchmarks beyond static single-turn settings toward interactive, sequential decision-making paradigms (Mohammadi et al., 2025).

Based on these observations, we argue that a comprehensive agent benchmark should reflect real deployments along three dimensions: task complexity (long-horizon, multi-step objectives), tool complexity (reasonable tool interfaces and coordinated tool use), and interaction complexity (diverse user behaviors and behavioral attributes)). Accordingly, evaluation should emphasize two central capabilities: (1) robust multi-turn instruction following with preference tracking under global constraints, and (2) long-horizon planning and reasoning with effective tool orchestration.

However, existing benchmarks still fall short. First, many focus on single-turn tasks (Li et al., 2025; Luo et al., 2025), or add multi-turn interaction without systematically modeling complex rule constraints (system- or user-level) that are essential in deployment (Liu et al., 2025b). Second, even in interaction-oriented benchmarks such as $\tau^2$-Bench (Barres et al., 2025), turn-level queries are often simple and solvable with only a few tool calls (often $< 3$), yielding shallow reasoning and short execution chains that under-

*Table 1.* Comparison of representative *user interaction benchmarks* and *travel planning benchmarks*. The table indicates whether each trait is fully addressed (✓), partially addressed (◐), or not addressed (✗). Detailed explanations for each trait are provided in Appendix A.

| Benchmark | Instruction Following | | Planning & Reasoning | | Task Complexity | Tool Complexity | | Interaction Complexity | | Scalable |
|---|---|---|---|---|---|---|---|---|---|---|
| | Constraint Adherence | Preference Alignment | Information Integration | Goal Management | Max Tool Calls & Avg Turns | Appro-priateness | Inter-Dependency | Behavior Attributes | Behavioral Diversity | Trainable |
| TravelPlanner (Xie et al., 2024) | ✗ | ◐ | ✓ | ◐ | [15,15] | ✗ | ✗ | ✗ | ✗ | ✗ |
| TripTailor (Wang et al., 2025) | ✗ | ✓ | ✓ | ◐ | [5,5] | ✗ | ✗ | ✗ | ✗ | ✓ |
| LLMs Get Lost (Laban et al., 2025) | ✗ | ✗ | ✗ | ✗ | [0,5] | N/A | N/A | ✗ | ✗ | ✗ |
| UserBench (Qian et al., 2025a) | ✗ | ✓ | ✓ | ✓ | [1,20] | ✗ | ✗ | ✗ | ◐ | ✓ |
| $\tau$-Bench (Yao et al., 2024) | ✓ | ✗ | ◐ | ✗ | [1,40] | ✓ | ✓ | ✗ | ✗ | ✓ |
| $\tau^2$-Bench (Barres et al., 2025) | ✓ | ✗ | ◐ | ✗ | [1,60] | ✓ | ✓ | ✓ | ✗ | ✓ |
| COMPASS (Qin et al., 2025) | ✗ | ✓ | ✓ | ◐ | [15,50] | ◐ | ✓ | ✓ | ◐ | ✗ |
| VitaBench (He et al., 2025) | ✗ | ✓ | ✓ | ✓ | [5,75] | ◐ | ✓ | ✓ | ✗ | ✗ |
| **TRIP-Bench(ours)** | ✓ | ✓ | ✓ | ✓ | [50,150] | ✓ | ✓ | ✓ | ✓ | ✓ |

represent long-horizon planning, iterative refinement, and error correction.

More importantly, benchmarks such as VitaBench (He et al., 2025) and COMPASS (Qin et al., 2025) often present instructions and context in segmented fragments. Even when intent ambiguity is introduced (e.g., intent-obfuscating rewrites in UserBench (Qian et al., 2025a)), prolonged interaction behaviors—such as revisions, rollbacks, and version control—remain under-modeled, limiting coverage of complex and dynamic real-world interaction patterns.

To address these limitations, we propose **TRIP-Bench**, a real-world benchmark based on travel planning that systematically evaluates agent capabilities along four dimensions: long-horizon **T**asks, complex **R**ules, diverse multi-turn **I**nteractions, and reasoning-driven **P**lanning.

TRIP-Bench is a large-scale travel-planning benchmark built on expanded and cleaned TripTailor data (Wang et al., 2025). It provides 18 tools and covers nearly 40 travel-need categories with 80+ natural-language formulations, enabling scalable multi-turn evaluation and training under complex constraints. Beyond simple "instruction sharding" (Laban et al., 2025), TRIP-Bench models nine categories of user behaviors and supports difficulty-controlled splits. The hard split includes four challenging interaction subsets—LIT, FIT, AIS, and PMR—capturing long dialogs, feasibility transitions, ambiguous intent, style shifts, and version control. In the hardest cases, dialogs can reach 15 turns with over **150** tool calls and total context beyond **200k** tokens, making TRIP-Bench a rigorous testbed for long-horizon planning, reasoning, and interaction robustness. Experiments show that most models score below 10% in strict mode, and even in loose mode, the best-performing model—GPT-5.2—reaches only 45%, posing a substantial challenge.

Beyond benchmark construction, we further propose **GTPO** (Group Relative Turn-level Policy Optimization), an online multi-turn reinforcement learning method to improve long-horizon interaction and reasoning. GTPO includes: (1) turn-level reward normalization, (2) global-instruction normalization, and (3) turn-level reward differencing, aiming to stabilize long-horizon decisions under dynamic interactions. Results on TRIP-Bench shows that GTPO-trained Qwen2.5-32B-Instruct outperforms the SFT model by over 10 pp under the loose setting and over 5 pp under the strict setting, and exceeds the base model by over 20 pp under the loose setting. Overall, our main contributions are as follows:

• We build a large-scale, tool-augmented simulation environment with modular data generation and validation, enabling scalable benchmarking and multi-turn training/evaluation under complex constraints and dynamic user behaviors.

• We conduct extensive experiments and in-depth analyses to systematically uncover the limitations of existing models under long-horizon reasoning, multi-turn diverse user behaviors, and global constraint adherence.

• We propose GTPO, an online multi-turn RL method that improves stable adherence to global rules and adaptation to dynamic preferences. After training with GTPO, Qwen2.5-32B-Instruct outperforms Gemini-3-Pro on TRIP-Bench.

## 2. Related Work

**Agent-user interaction benchmarks.** Agent benchmarks have evolved from early single-turn, single-step tool-use settings (Huang et al., 2023; Qin et al., 2023) to more challenging single-turn, multi-step benchmarks such as TravelPlanner (Xie et al., 2024) and MCP Universe (Luo et al., 2025). Multi-turn interactive benchmarks (e.g., ToolTalk (Farn & Shin, 2023) and BFCL (Patil et al., 2025)) further support conversational tool execution, but their reliance on predefined dialogue trajectories limits agent autonomy and behavioral diversity. UserBench (Qian et al., 2025a) introduces intention ambiguity yet focuses on relatively simple tasks. Broader frameworks (e.g., $\tau^2$-Bench) evaluate instruction-following under verbose policy constraints in online customer-service environments. Recent efforts such as VitaBench (He et al., 2025) and COMPASS (Qin et al., 2025) enhance specific aspects, including tool diversity, dynamic interactions, and longer horizons. Still, no existing

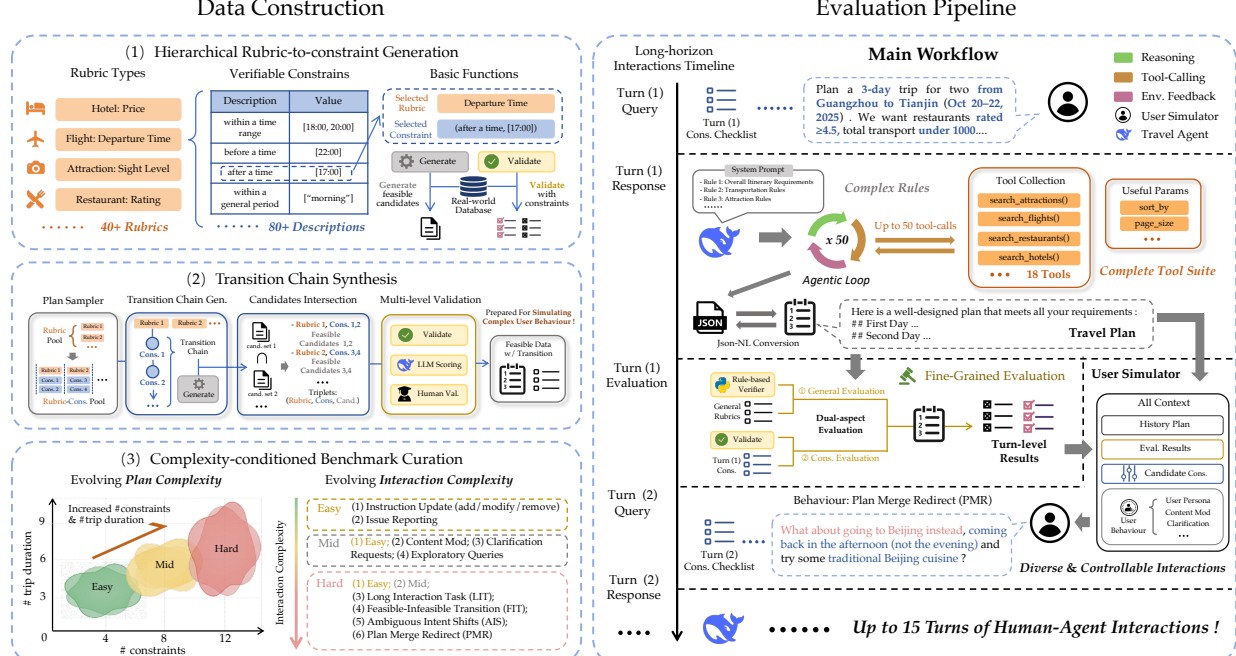

*Figure 1.* Overview of TRIP-Bench. **Left:** data construction via rubric-to-constraint generation, progressive modification-chain synthesis, and complexity-conditioned task curation. **Right:** long-horizon evaluation pipeline where a travel agent iteratively plans with a unified suite of tools and is assessed by rule-based and turn-level metrics under diverse user-simulator interactions.

benchmark provides a unified evaluation that simultaneously stresses complex instruction-following, long-horizon reasoning, and diverse user–agent interaction behaviors. TRIP-bench addresses this gap by introducing long-horizon tasks with complex rules and rich multi-turn interactions requiring spatiotemporal planning and reasoning.

**Multi-turn Reinforcement Learning.** Tool use is increasingly studied through reinforcement learning with outcome-driven rewards, enabling agents to autonomously explore and improve tool-calling strategies in interactive environments (Jin et al., 2025; Singh et al., 2025). However, most work targets single-turn multi-step tasks, treating the user query as fixed context and optimizing primarily for a single response (Xue et al., 2025; Xi et al., 2025). For multi-turn dialogue, a common simplification concatenates prior turns into a long context, ignoring the distributional shift in conversation histories as the policy changes. REFUEL (Gao et al., 2024) shows this induces covariate shift: training uses static offline histories, whereas deployment observes histories generated by the evolving policy, with mismatch compounding over turns. Methods such as MUA-RL (Zhao et al., 2025) and UserRL (Qian et al., 2025b) mitigate this by integrating dynamic user simulation into the RL loop to optimize for genuine multi-turn interactions. Nonetheless, these approaches largely focus on direct user–LLM dialogue rather than long-chain tool invocation, leaving multi-turn tool use in dynamic settings an open challenge.

## 3. TRIP-Bench

### 3.1. Environment and Tools

We extend the Triptailor (Wang et al., 2025) dataset by enriching POI attributes (e.g., hotel room types, restaurant set menus) and fixing formatting, logic, and consistency issues to support reliable task generation and evaluation. The final dataset covers 40 cities with 6k+ attractions, 80k+ hotels, 400k+ restaurants, and 1M+ distinct products.

We further build a unified tool interface and implement 18 tools for transportation, attractions, restaurants, hotels, and general utilities. Tools provide field-based filtering, sorting, and result-size control, enabling systematic evaluation of tool invocation, constraint understanding, and compositional decision-making. See Appendix B.1 for details.

### 3.2. Task Synthesis

**Meta-information Synthesis.** We sample all two- and three-city combinations among the 40 cities and generate candidate itineraries by assigning distance-based stays (2–7 days) and sampling departure dates and group sizes. Transportation tools are used to filter out infeasible candidates. For three-city cases, we retain only itineraries where either two cities are within 500 km or the three cities are roughly collinear, matching typical travel routes. The resulting seeds include ∼6k two-city and ∼4k three-city itineraries, which define the itinerary meta-information.

**Rubric and Constraint Construction.** We collect approximately 40 common requirement categories from real-world travel planning scenarios and curate over 80 diverse natural language expressions. For each expression $e$, we define two paired functions: the generator $G(e)$ (i.e., generate), which produces the fine-grained selection range $R$ and the corresponding feasible ID set $\mathcal{F}$ ( Figure 1, left (1)), and the validator $V(e, i)$ (i.e., validate), which checks whether a given single ID $i$ satisfies the expression. The complete rubric, detailed explanations, and additional examples are provided in Appendix B.2.

**Modification Chain Construction.** As shown in Figure 1, left (2), we provide meta-information and a rubric-specific candidate set, and prompt the model to generate a modification chain of up to three steps that becomes progressively more restrictive, mirroring iterative user refinement. To reduce redundancy—cases where earlier constraints already entail later ones—we use two strategies: (1) trajectory-based trimming, which samples trajectories and checks whether earlier constraints satisfy later ones, discarding the prefix of the chain and retaining only its suffix; and (2) rubric-level adjustment, which shortens the target chain length for rubrics prone to redundancy. This increases the share of chains in which each new constraint induces a substantive change, while preserving some redundancy for realism.

**Task Generation.** As shown in Figure 1, left (3), we first partition tasks into three difficulty tiers (easy, mid, hard) based on trip length (days), number of cities, number of constraints, and the difficulty of simulated user behaviors. Detailed criteria are given in Appendix B.3. We then sample rubrics along four dimensions—transportation, attractions, restaurants, and hotels—targeting an approximately uniform distribution over rubric counts. For each sampled rubric, we set the number of modification steps to match the desired modification-chain length. Given a set of constraint expressions $\mathcal{E}$, we derive an initially filtered candidate set $\mathcal{C}_0 = \bigcap_{e \in \mathcal{E}} \mathcal{F}_e$, where $(R_e, \mathcal{F}_e) = G(e)$. Because subset-/containment constraints (e.g., "must include one or more restaurants of certain types") may not fully filter candidates or certify solvability, we further verify feasibility by defining $\mathcal{C} = \{i \in \mathcal{C}_0 : \forall e \in \mathcal{E}, V(e, i) = 1\}$. Finally, we require $|\mathcal{C}| \geq (4\text{--}10)\times$ the trip length (days) to ensure sufficient flexibility for itinerary construction. This produces a base task set spanning all difficulty levels. Building on the hard subset, we combine user behaviors to create four more challenging evaluation sets:

**LIT (Long Interaction Task):** LIT increases dialogue turns by using fewer initial constraints and smaller per-turn updates in the user simulator.

**FIT (Feasible–Infeasible Transition):** FIT selects chains that are infeasible at the current step but become feasible after rolling back one step (equivalent to deletion when

the chain length is 1). It then composes 2–4 infeasible requirements (thus requiring 2–4 rollbacks) and dynamically injects rollback instructions during execution—when the agent declares infeasibility, at simulator-chosen moments, or at the end—so that the final requirements are feasible.

**AIS (Ambiguous Intent Shifts):** AIS introduces ambiguous constraints throughout the dialogue and reveals explicit preferences/corrections only when the model errs or proactively asks clarifying questions, using five interaction styles to better capture user interaction patterns.

**PMR (Plan Merge Redirect):** PMR constructs two similar itineraries that share 6–9 modification chains but differ elsewhere, and inserts trigger nodes that prompt the simulator to switch between itineraries, optionally roll back after several turns, or merge the two plans.

### 3.3. Quality Control

Although each component in our pipeline is solvable on its own, combining them can produce unrealistic cases (e.g., preference–budget mismatches). We address this with prompt-based model scoring plus manual review. Because travel plans have spatiotemporal dependencies—local feasibility doesn't ensure global feasibility—we sample full plans, evaluate them, and manually check whether flagged issues are repairable. This two-stage validation keeps tasks practical and globally feasible.

### 3.4. User Simulation

Unlike VitaBench (He et al., 2025) and UserBench (Qian et al., 2025a), which provide a full instruction block and let the model respond freely, or COMPASS (Qin et al., 2025), which relies on fully predefined scripts where the model only renders dialogue style, we introduce a user dialogue graph and maintain a per-turn list of active user preferences. At each turn, we dynamically update the user simulator prompt and vary behavioral diversity based on the difficulty level. We constrain the update pipeline by ensuring later changes are not visible to earlier steps, and by switching preferences only at a small set of key nodes. This balances autonomy, diversity, and controllability in user simulation.

### 3.5. Evaluation metrics

We define 12 general constraints for the task: 4 for *basic feasibility* and 8 for *planning soundness* (see Appendix B.4). As user queries contain varying numbers of user constraints, we report two aggregate metrics: $\text{Overall}_{\text{Strict}} = \mathbb{I}(F_{\text{feas}} = 0 \wedge F_{\text{sound}} = 0 \wedge F_{\text{user}} = 0)$ and $\text{Overall}_{\text{Loose}} = \mathbb{I}(F_{\text{feas}} = 0 \wedge F_{\text{sound}} \leq 2 \wedge F_{\text{user}} \leq 1)$, where the loose criterion keeps feasibility strict while allowing up to two soundness violations and one user-constraint violation; $F_{\text{feas}}$, $F_{\text{sound}}$, and $F_{\text{user}}$ denote the numbers of violated feasibility,

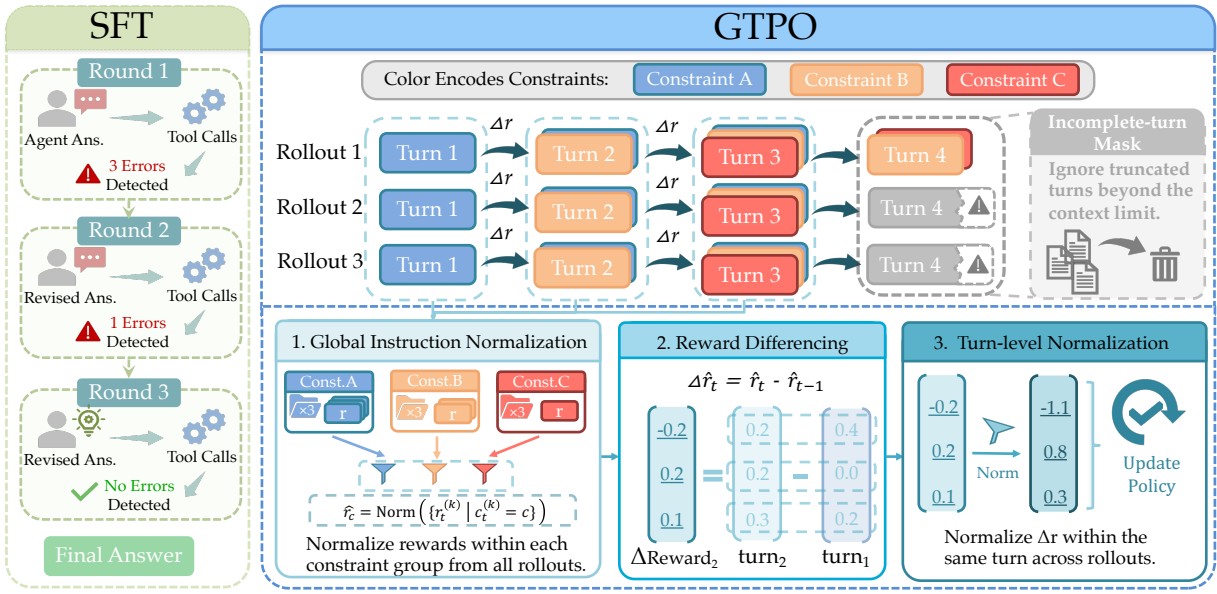

Figure 2. Overview of our training pipeline. **Left:** 120k trajectories are sampled from synthesized prompts, repaired with three rounds of error feedback, and filtered to obtain high-quality rollouts for SFT. **Right:** GTPO optimizes on groups of multi-turn rollouts by (i) global instruction-wise normalization, (ii) turn-wise reward differencing, and (iii) per-turn reward normalization.

soundness, and user constraints, respectively.

## 4. Method

### 4.1. Data Construction

**SFT.** We synthesize $\sim$120k samples for SFT sampling using *DeepSeek-v3.2 (no-think)*. Due to the task difficulty, only $\sim$500 fully correct trajectories are obtained. Notably, many incorrect trajectories still exhibit reasonable tool use and coherent reasoning. We therefore fix the tools and outputs and provide only error feedback for three rounds of repair, producing $\sim$9k trajectories that pass evaluation (Figure 2, left). To mitigate potential hacking risks, we further keep only trajectories with full reasoning and planning scores, yielding $\sim$3k trajectories, which are combined with Toucan (Xu et al., 2025) data for SFT cold-start training.

**RL.** We sample once over the same 120k inputs using the SFT-trained model and retain trajectories that satisfy a relaxed criterion, $F_{\text{feas}} = 0 \wedge (F_{\text{sound}} + F_{\text{cons}} \leq 5)$, yielding 7,040 samples for RL training.

### 4.2. GTPO: Group Turn-level Preference Optimization

#### 4.2.1. PRELIMINARY

Given a dialogue context (prompt) $x$, we sample a *group* of multi-turn rollouts from the current policy $\pi_\theta$:

$$\tau^{(k)} = \{(u_1, a_1^{(k)}), \ldots, (u_{T_k}, a_{T_k}^{(k)})\}, \quad k = 1, \ldots, K, \tag{1}$$

where $u_t$ and $a_t^{(k)}$ are the user input and assistant response at turn $t$, and $T_k$ is the number of turns in rollout $k$.

**Raw Reward.** At turn $t$, each constraint $i \in \mathcal{I}_t$ yields a binary score $c_{t,i}^{(k)} \in \{0, 1\}$, and basic feasibility is a hard gate $\mathbb{I}_{\text{feas}}^{(k,t)} = \mathbb{I}(F_{\text{feas}}^{(k,t)} = 0)$. The (pre-GTPO) turn reward is

$$r_{t,\text{raw}}^{(k)} = \mathbb{I}_{\text{feas}}^{(k,t)} \cdot \frac{1}{|\mathcal{I}_t|} \sum_{i \in \mathcal{I}_t} c_{t,i}^{(k)}. \tag{2}$$

#### 4.2.2. KEY COMPONENTS OF GTPO

**Turn-wise Reward Differencing.** As later turns are strongly influenced by earlier turns, a slightly worse turn may still receive a higher reward due to inherited structure. To emphasize relative improvement, we apply turn-wise reward differencing $d_t^{(k)}$, which is defined as:

$$d_t^{(k)} = \begin{cases} r_1^{(k)}, & t = 1, \\ r_t^{(k)} - r_{t-1}^{(k)}, & t \geq 2. \end{cases} \tag{3}$$

If turn $(t-1)$ is infeasible, we subtract $r_{t-1}^{\max}$ within the same group instead of $r_{t-1}^{(k)}$, where

$$r_{t-1}^{\max} = \max_{k' \in \mathcal{K}_{t-1}} r_{t-1}^{(k')}. \tag{4}$$

Thus, for $t \geq 2$,

$$d_t^{(k)} = r_t^{(k)} - \mathbb{I}_{\text{feas}}^{(k,t-1)} r_{t-1}^{(k)} - \left(1 - \mathbb{I}_{\text{feas}}^{(k,t-1)}\right) r_{t-1}^{\max}. \tag{5}$$

*Table 2.* Performance comparison of different models under loose and strict evaluation across difficulty levels.

| Models | Easy | | Mid | | Hard LIT | | Hard FIT | | Hard AIS | | Hard PMR | | Overall | |
|---|---|---|---|---|---|---|---|---|---|---|---|---|---|---|
| | loose | strict | loose | strict | loose | strict | loose | strict | loose | strict | loose | strict | loose | strict |
| *Non-thinking Models* | | | | | | | | | | | | | | |
| Kimi-K2-0905-Preview | 13.0 | 0.0 | 0.0 | 0.0 | 0.0 | 0.0 | 0.0 | 0.0 | 0.0 | 0.0 | 0.0 | 0.0 | 3.3 | 0.0 |
| Qwen3-235B-A22B-Instruct-2507 | 16.0 | 2.0 | 5.0 | 0.0 | 0.0 | 0.0 | 2.0 | 0.0 | 2.0 | 0.0 | 0.0 | 0.0 | 5.8 | 0.5 |
| GPT-5.2 (w/o thinking) | 24.0 | 2.0 | 14.0 | 0.0 | 10.0 | 0.0 | 6.0 | 0.0 | 8.0 | 0.0 | 6.0 | 0.0 | 13.3 | 0.5 |
| GLM-4.7 (w/o thinking) | 34.0 | 0.0 | **20.0** | 0.0 | 6.0 | 0.0 | 0.0 | 0.0 | 0.0 | 0.0 | 0.0 | 0.0 | 14.8 | 0.0 |
| Claude-Sonnet-4.5 (w/o thinking) | 36.0 | 7.0 | 18.0 | 0.0 | 10.0 | 0.0 | 6.0 | 0.0 | 10.0 | 0.0 | 4.0 | 0.0 | 17.3 | 1.8 |
| Gemini-3-Flash (w/o thinking) | 36.0 | **22.0** | 11.0 | 0.0 | 8.0 | 0.0 | 6.0 | 0.0 | **16.0** | 0.0 | **14.0** | 0.0 | 17.3 | **5.5** |
| Gemini-3-Pro (w/o thinking) | **44.0** | 12.0 | 9.0 | 0.0 | 12.0 | 0.0 | 4.0 | 0.0 | 12.0 | 0.0 | 10.0 | 0.0 | 18.0 | 3.0 |
| DeepSeek-V3.2 (w/o thinking) | 39.0 | 5.0 | **20.0** | 3.0 | 16.0 | 2.0 | 8.0 | 0.0 | 2.0 | 0.0 | 4.0 | 0.0 | **18.5** | 2.3 |
| *Thinking Models* | | | | | | | | | | | | | | |
| Qwen3-235B-A22B-Thinking-2507 | 0.0 | 0.0 | 0.0 | 0.0 | 0.0 | 0.0 | 0.0 | 0.0 | 0.0 | 0.0 | 0.0 | 0.0 | 0.0 | 0.0 |
| Kimi-K2-Thinking | 35.0 | 5.0 | 8.0 | 4.0 | 0.0 | 0.0 | 0.0 | 0.0 | 0.0 | 0.0 | 0.0 | 0.0 | 10.8 | 2.3 |
| Gemini-3-Pro (w/ thinking) | 42.0 | 11.0 | 16.0 | 0.0 | 16.0 | 0.0 | 0.0 | 0.0 | 18.0 | 0.0 | 10.0 | 0.0 | 20.0 | 2.8 |
| GLM-4.7 (w/ thinking) | 38.0 | 16.0 | 29.0 | 0.0 | 10.0 | 0.0 | 0.0 | 0.0 | 0.0 | 0.0 | 18.0 | 0.0 | 20.3 | 4.0 |
| Gemini-3-Flash (w/ thinking) | 44.0 | 25.0 | 25.0 | 0.0 | 10.0 | 0.0 | 0.0 | 0.0 | **26.0** | 0.0 | 12.0 | 0.0 | 23.3 | 6.3 |
| Claude-Sonnet-4.5 (w/ thinking) | 58.0 | 27.0 | 31.0 | 6.0 | 28.0 | 0.0 | 10.0 | 0.0 | 22.0 | 0.0 | 18.0 | 2.0 | 32.0 | 8.5 |
| DeepSeek-V3.2 (w/ thinking) | **71.0** | 31.0 | 41.0 | 9.0 | 36.0 | 2.0 | 14.0 | 0.0 | **26.0** | 0.0 | 20.0 | 2.0 | 40.0 | 10.5 |
| GPT-5.2 (w/ thinking) | 66.0 | **49.0** | **55.0** | **13.0** | **44.0** | **14.0** | **18.0** | 0.0 | 20.0 | 0.0 | **36.0** | **10.0** | **45.0** | **18.5** |

**Turn-level Reward Normalization.** For each turn $t$, let $\mathcal{K}_t$ be the set of rollouts that complete and are evaluable at turn $t$. To stabilize per-turn normalization statistics, we only normalize turns with sufficient samples, requiring $|\mathcal{K}_t| \geq K/2$, and mask turns that exceed the context budget in the loss. We normalize $\{d_t^{(k)}\}_{k \in \mathcal{K}_t}$ across the group via z-score:

$$\mu_t = \frac{1}{|\mathcal{K}_t|} \sum_{k \in \mathcal{K}_t} d_t^{(k)}, \qquad \sigma_t = \sqrt{\frac{1}{|\mathcal{K}_t|} \sum_{k \in \mathcal{K}_t} \left(d_t^{(k)} - \mu_t\right)^2}.$$
(6)

The turn-level advantage is then

$$A_t^{(k)} = \frac{d_t^{(k)} - \mu_t}{\sigma_t + \epsilon}, \qquad k \in \mathcal{K}_t.$$
(7)

Importantly, $A_t^{(k)}$ is turn-local: each turn has its own advantage and advantages do not propagate across turns.

### 4.2.3. FINAL OBJECTIVE

We optimize the policy parameters $\theta$ by maximizing the following GTPO objective:

$$J_{\text{GTPO}}(\theta) = \mathbb{E}_{x, \{\tau^{(k)}\}_{k=1}^K} \left[ \frac{1}{K} \sum_{k=1}^K \frac{1}{T_k} \sum_{t=1}^{T_k} \frac{1}{L_{k,t}} \sum_{j=1}^{L_{k,t}} m_{t,j}^{(k)} \right.$$

$$\left( \min \left( \rho_{t,j}^{(k)}(\theta) A_t^{(k)}, \text{clip}\left(\rho_{t,j}^{(k)}(\theta), 1 - \epsilon, \right. \right. \right.$$

$$\left. \left. \left. 1 + \epsilon\right) A_t^{(k)}\right) - \beta D_{\text{KL}}\left(\pi_\theta \| \pi_{\text{ref}}; h_{t,j}^{(k)}\right)\right) \right].$$
(8)

where $m_{t,j}^{(k)} \in \{0, 1\}$ is the token-level mask, $A_t^{(k)}$ is the turn-level advantage defined in Eq. (7), and $\rho_{t,j}^{(k)}(\theta) = \pi_\theta(a_{t,j}^{(k)} \mid h_{t,j}^{(k)}) / \pi_{\theta_{\text{old}}}(a_{t,j}^{(k)} \mid h_{t,j}^{(k)})$ is the PPO importance ratio, with $\text{clip}(\cdot, 1 - \epsilon, 1 + \epsilon)$ using threshold $\epsilon$. $D_{\text{KL}}(\pi_\theta \| \pi_{\text{ref}}; h_{t,j}^{(k)})$ denotes the per-token KL divergence to the reference policy $\pi_{\text{ref}}$, weighted by $\beta$.

## 5. Experiments

### 5.1. Settings

**Models.** We evaluate a collection of recent large language models, including Kimi-K2 (Bai et al., 2025), Qwen3-235B-A22B (Yang et al., 2025), GLM-4.7, DeepSeek-V3.2 (Liu et al., 2025a), Gemini-3 (Flash and Pro), GPT-5.2, and Claude-Sonnet-4.5. For models that support different reasoning behaviors, we conduct evaluations under two configurations: with explicit reasoning disabled and with the default reasoning strength enabled.

**Implementation Details.** During evaluation, all models use their default temperature settings; when unspecified, the temperature is set to 0.7. We employ DeepSeek-V3.2 as the user simulator with a temperature of 0.7. Training is conducted on Qwen2.5-14B-Instruct and Qwen2.5-32B-Instruct. For the trained models, evaluation is performed only on the easy and mid subsets, as the hard subset often requires context lengths exceeding 128k tokens, which is beyond the maximum context length supported by the models. Additional training details are provided in Appendix D.1.

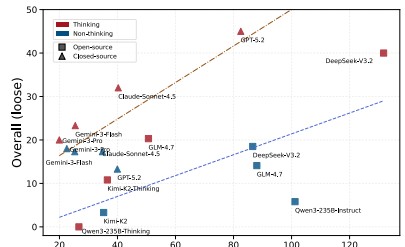 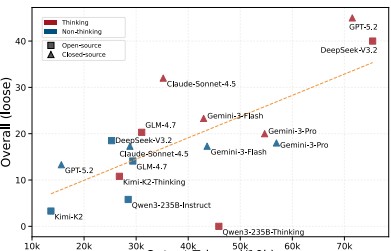 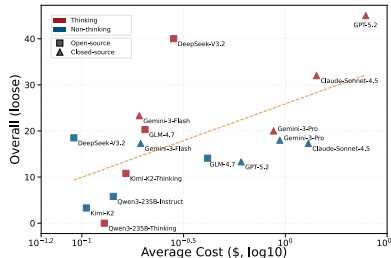

*Figure 3.* **Left:** Performance vs resource use. Three scatter plots: performance vs #turns (left), output tokens per 10k (middle), and avg reasoning cost (USD, log; right). Models: thinking vs non-thinking; open- vs closed-source by marker shape; dashed lines are trend fits.

## 5.2. Main Results

**TRIP-bench poses a significant challenge.** As shown in Table 2, TRIP-bench is highly challenging under both strict and loose evaluation. Under the strict metric, performance is extremely poor even on the Easy subset, with many models near zero accuracy and the best score reaching only 18.5. This difficulty persists under relaxed evaluation, where the highest loose score is still limited to 45, indicating that errors arise not only from strict verification but from fundamental reasoning limitations. These results highlight two key challenges. First, TRIP-bench requires long-horizon, multi-constraint reasoning, where satisfying individual constraints is insufficient. Second, performance degrades sharply on more difficult behavioral subsets: FIT remains unsolved under strict evaluation, and PMR shows only marginal improvements while consistently lagging behind LIT. Overall, the results reveal substantial limitations of current models in handling complex and behaviorally demanding scenarios, even under relaxed criteria.

**Thinking dramatically improves performance under both loose and strict evaluation, but remains insufficient for fully satisfying Hard cases.** Enabling thinking yields consistent and substantial gains across easier splits, improving both accuracy and robustness. For instance, on Easy-strict, performance rises from 5.0 to 31.0—an absolute gain of 26 percentage points—and the Overall-strict score increases from 2.3 to 10.5. Similar improvements are also observed on the Hard sets under loose evaluation (often by more than 15 points), suggesting that reasoning augmentation helps models reach partially correct or near-complete solutions. However, Hard-strict performance remains uniformly low across models, indicating that current thinking mechanisms still fall short of producing comprehensive, fully correct, and verifiable outcomes when faced with the most challenging user behaviors and strict checking.

**GTPO better aligns training with dynamic user interaction and multi-constraint reasoning, with gains that scale to stronger models.** As shown in Table 3, GTPO consistently outperforms SFT and GRPO under both loose and strict evaluation, yielding more stable and balanced improvements. When scaled to Qwen2.5-32B-Instruct, GTPO

*Table 3.* Performance on Easy/Mid. ST denotes single-turn training; MT denotes multi-turn training where only the final-turn reward is used. In GTPO, we ablate three key components: Global Instruction Normalization (GIN), Turn-wise Reward Differencing (TRD), and Turn-level Reward Normalization (TRN) (see §4.2.2). *GTPO (w/o X)* removes component(s) X from the full setup.

| Models | Easy | | Mid | |
|---|---|---|---|---|
| | *loose* | *strict* | *loose* | *strict* |
| *Base: Qwen2.5-14B-Instruct* | | | | |
| Base | 0 | 0 | 0 | 0 |
| +SFT | 16 | 4 | 8 | 0 |
| +GRPO (ST) | 29 | 0 | 12 | 0 |
| +GRPO (MT) | 30 | 4 | 16 | 0 |
| +GTPO (w/o GIN, TRD) | 32 | 12 | 16 | 0 |
| +GTPO (w/o TRD) | 34 | 10 | **20** | 0 |
| +GTPO (full) | **35** | **13** | 18 | 0 |
| *Base: Qwen2.5-32B-Instruct* | | | | |
| Base | 0 | 0 | 0 | 0 |
| +SFT | 32 | 3 | 5 | 0 |
| +GTPO (full) | **49** | **21** | **40** | **5** |

substantially strengthens performance on harder settings, achieving 40 on Mid-loose and 21 on Easy-strict, and notably surpassing Gemini-3 Pro under the same evaluation. These results indicate that GTPO provides a more effective training signal for long-horizon, multi-constraint reasoning under dynamic user interaction. Our training curves are shown in Figure 6.

## 6. In-Depth Analysis

**Model performance improvements increasingly depend on deeper reasoning and longer token generation, but these gains incur substantial marginal costs.** As illustrated in Fig. 3, performance scales approximately linearly with the number of output tokens. However, due to the additional input-token overhead introduced by multi-step reasoning, the relationship between performance gains and inference cost is closer to logarithmic. Notably, DeepSeek-V3.2 Thinking demonstrates exceptional cost-effectiveness: under the Loose evaluation regime, it achieves performance

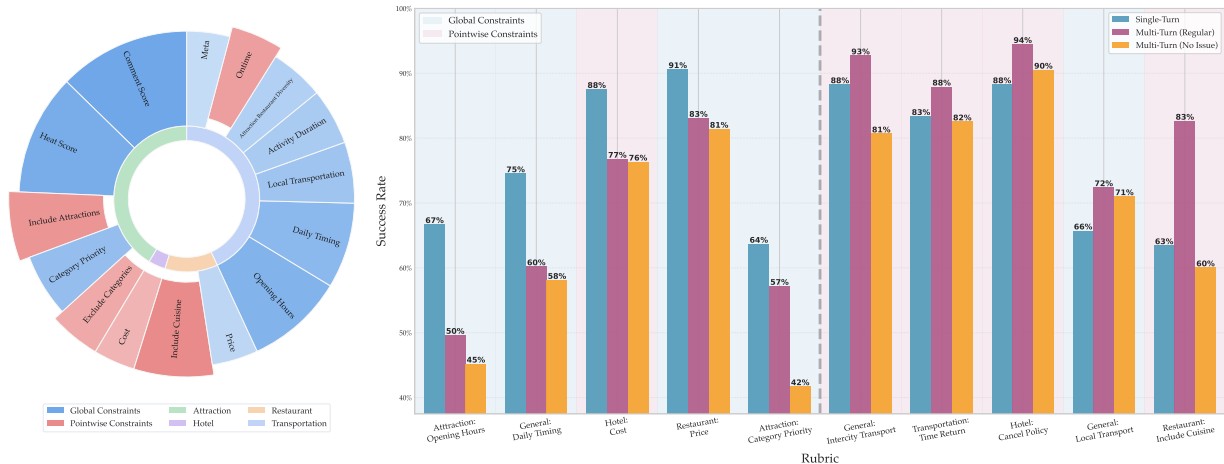

*Figure 4.* **Left:** Breakdown of the top-15 highest-error constraints by domain and constraint type (Global vs. Pointwise) in the multi-turn setting. **Right:** DeepSeek-V3.2-Thinking score rates per rubric under single-turn and multi-turn (regular vs. no-issue) settings.

comparable to GPT-5.2 at only about 10% of the cost (approximately $0.25). Although a clear gap remains under the stricter Strict metric, this result indicates that for error-tolerant applications, DeepSeek-V3.2 offers a highly economical alternative without pursuing peak performance at all costs. Finally, aside from the limited catch-up trend exhibited by DeepSeek-V3.2, a pronounced generational gap persists between current open-source models and leading closed-source systems, with open-source models maintaining a significantly lower overall performance baseline.

**Multi-turn interactions often degrade on complex tasks as global consistency gradually erodes, whereas single-turn interactions more reliably satisfy strict global constraints in a single pass.** As shown in Fig. 4, under strong global constraints—such as validating POI opening hours or optimizing attraction sequences—single-turn accuracy exceeds multi-turn by roughly 10 percentage points on average. Together with Table 5, this gap widens substantially in the Hard subset: on the two most challenging evaluation sets, multi-turn performance under the Loose metric falls more than 20 percentage points behind single-turn. These results suggest that complex user behaviors coupled with strong global constraints make multi-turn systems more prone to global mismatch. In contrast, multi-turn interactions remain competitive on local constraints (e.g., hotel cancellation policies or specific cuisine requirements), which change infrequently and can be corrected via feedback. By incrementally incorporating constraints and rectifying errors, multi-turn refinement enables progressive convergence, outperforming single-turn on localized, point-specific tasks.

**Ablation Study of GTPO.** Table 3 reveals the impact of each GTPO component. While Turn-level Reward Normalization (TRN) stabilizes training and improve performance, the lack of global constraint balancing and turn-by-turn objectives limits its performance. Adding Global Instruction

Normalization (GIN) improves average rewards by calibrating constraint satisfaction across dialogue history, though the "reward inheritance" issue continues to hinder the complete pass rate (strict metric). Finally, Turn Reward Differencing (TRD) yields the best results; by using the previous turn's reward as a baseline, it prioritizes incremental gains, substantially boosting the complete pass rate while maintaining high overall rewards.

**Reliability of the user simulator.** We conducted a manual evaluation of the user simulator. We randomly sampled 20 trajectories (104 turns in total) and verified whether the issued instruction ID was consistently reflected in subsequent queries, achieving 98% reliability. In addition, we sampled 10 trajectories in AIS (62 turns total) and rated the simulator on whether the intended ambiguity was properly captured and whether the style simulation was faithful, obtaining an average score of 4.7/5. See Appendix E for details.

**Exploration vs. reliability.** As shown in Figure 5, pass@k increases with more samples, indicating non-trivial exploratory ability, but pass@1 remains low and strict evaluation is substantially worse, highlighting limited single-try reliability under hard constraints. Meanwhile, $avg^k$ stays stable across $k$, suggesting the stability of our bench.

## 7. Conclusion

We present TRIP-Bench, a long-horizon interactive benchmark for realistic travel planning. It tests global constraint adherence, long-term planning, multi-tool use, and dynamic user behavior. Results show even strong models struggle—especially on harder interactions—highlighting gaps in cross-turn consistency and meeting global constraints. We also propose GTPO, an online multi-turn RL method with global instruction normalization and turn-level reward shaping, delivering consistent gains over strong baselines.

## Impact Statement

This work is sponsored by Beijing Nova Program. TRIP-Bench is introduced to advance research on reliable long-horizon interactive agents, especially for settings that require maintaining global constraints, coordinating multi-tool execution, and adapting to evolving user preferences over many turns. Parts of the environment are constructed from publicly available data (e.g., travel-related information), and any included content does not represent the authors' viewpoints. To foster reproducible progress, we plan to release the benchmark, evaluation scripts, and supporting resources, and we will also release our trained models to facilitate follow-up research in training and evaluation.

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

# A. Detailed Explanations for Comparison Traits

- **Constraint Adherence**: Strictly comply with all constraints explicitly specified in the system prompt, including explicit rule requirements, special-case handling logic, system boundary conditions, and operational/runtime limitations.

- **Preference Alignment**: Accurately satisfy the user's explicitly stated needs while maintaining consistency and continuity of preferences across multi-turn interactions. When user preferences change or conflict, identify the discrepancy, reconcile it, and resolve it reasonably to avoid inconsistent responses.

- **Information Integration**: Integrate temporal, spatial, and commonsense information, and incorporate environmental context and state changes to reason systematically about the problem, producing solutions that are logically consistent, context-aware, and aligned with the current state.

- **Goal Management**: When user goals are unclear or information is incomplete, progressively refine vague requirements into clear, executable, and verifiable task objectives through clarifying questions, contextual inference, and iterative reasoning. In complex tasks, handle multiple interdependent goals and constraints (e.g., cost, time, risk, resources) simultaneously by balancing trade-offs, coordinating dependencies, and dynamically adjusting plans to achieve a globally optimal or satisfactory solution rather than a locally optimal one.

- **Execution Complexity**: Tasks may require multi-step execution, persistent state tracking, or conditional branching decisions, increasing the depth of planning and execution. Complexity can be reflected by observable indicators such as the maximum number of tool calls within a single reasoning step and the average number of tool calls across the overall interaction; these indicators also capture execution overhead and the risk of error accumulation.

- **Appropriateness**: Tool usage should maintain clear functional boundaries and consistent semantic definitions. Outputs should focus on core information and remain low-noise, using appropriate parameters and filtering mechanisms to reduce redundancy rather than accumulating irrelevant content in context.

- **Inter-Dependency**: When tools exhibit sequential dependencies, result passing, or cascading invocation relationships, the depth of reasoning and orchestration complexity increases significantly, placing higher demands on overall planning capability and state management.

- **Behavior Attributes**: Model different users' behavioral attributes, including emotional states (e.g., impatience, anxiety), interaction styles (e.g., detail-oriented, model-reliant), and engagement levels that change with model performance (e.g., reduced willingness to respond after repeatedly receiving similar answers).

- **Behavioral Diversity**: Cover a wide range of realistic user behavior changes, including adding, modifying, deleting, or reverting instructions during the conversation; redirecting intent; merging multiple goals; making partial edits to model-generated content; pointing out model errors; requesting further clarification; or seeking advice in an exploratory manner—demonstrating the system's ability to adapt to complex and dynamic interactive behaviors.

# B. TRIP-Bench Construction

## B.1. Environment and Tools

### B.1.1. ATTRACTION TOOLS

**func: search_attractions**
DESCRIPTION: Search attractions in a city with flexible filtering, ranking, and pagination support.
PARAMETERS: city (str), attraction_name (opt), categories (opt), longitude/latitude (opt), distance_threshold (opt), rating (opt), sight_level (opt), comment_count (opt), free_only (bool), sort_key (opt), sort_order (opt), page, page_size.
RETURNS: A ranked list of attractions with metadata (location, rating, popularity, opening hours, price, distance).

**func: get_attraction_detail_with_products**
DESCRIPTION: Retrieve detailed information of a specific attraction and its ticket products.
PARAMETERS: poi_id (str).
RETURNS: Full attraction profile including categories, ratings, opening hours, features, and purchasable tickets.

**func: get_attraction_coordinates**
DESCRIPTION: Obtain geographic coordinates of a given attraction.

PARAMETERS: poi_id (str).
RETURNS: Latitude and longitude of the specified attraction.

### B.1.2. HOTEL TOOLS

**func: search_hotels**
DESCRIPTION: Search hotels in a specified city for a stay window, with multi-criteria filtering (price, distance, ratings, amenities, room types, cancellation policy) plus sorting and pagination.
PARAMETERS: city (str), check_in_date (YYYY-MM-DD), check_out_date (YYYY-MM-DD), price_min/max (opt), longitude/latitude (opt), distance_threshold (opt), hotel_type (opt), stars (opt), review_count (opt), good_remarks_rate (opt), product/environment/service_rating (opt), room_types (opt), cancel_policy (opt), is_pet_friendly (opt), has_breakfast (opt), sort_key (opt), sort_order (opt), page, page_size.
RETURNS: A ranked list of hotels with key metadata (type, price, rating, review count, coordinates, distance).

**func: get_hotel_detail_with_products**
DESCRIPTION: Retrieve a hotel profile and its bookable room products for given dates, with product-level filtering and affordability/occupancy-aware ranking.
PARAMETERS: hotel_id (str), check_in_date (YYYY-MM-DD), check_out_date (YYYY-MM-DD), room_num (opt), person_num (opt), room_type (opt), min_breakfast_per_room (opt), cancel_policy (opt), has_window (opt), page, page_size.
RETURNS: Hotel summary followed by paginated product lines (room type, occupancy, breakfast, cancellation, window, nightly price); products that do not satisfy occupancy/room-count constraints are clearly separated.

**func: get_hotel_coordinates**
DESCRIPTION: Obtain geographic coordinates of a hotel by its ID.
PARAMETERS: hotel_id (str).
RETURNS: Latitude and longitude of the specified hotel, or a failure message if not found.

### B.1.3. FLIGHT TOOLS

**func: search_flights**
DESCRIPTION: Search available flights between two cities on a specific date with time-window filtering, sorting, and pagination.
PARAMETERS: departure_city (str), arrival_city (str), date (YYYY-MM-DD), dep_period (opt), arr_period (opt), sort_key (opt: time/price), sort_order (opt), page, page_size.
RETURNS: A ranked list of flight options with schedule, airline, airports, and minimum available price.

**func: get_flight_detail_with_products**
DESCRIPTION: Retrieve detailed flight information and purchasable ticket products for a given date.
PARAMETERS: flight_id (str), date (YYYY-MM-DD), source_platform (opt), seat_type (opt).
RETURNS: Flight summary with punctuality statistics, followed by available ticket products including platform, seat class, and price.

**func: get_airport_coordinates**
DESCRIPTION: Obtain geographic coordinates of an airport via exact or fuzzy name matching.
PARAMETERS: airport_name (str).
RETURNS: Latitude and longitude of the matched airport, or a failure message if not found.

### B.1.4. TRAIN TOOLS

**func: search_trains**
DESCRIPTION: Search trains between a departure/arrival city pair with time-window filtering, optional price/time ranking, and pagination (accelerated via a pre-built (dep, arr) index).
PARAMETERS: departure_city (str), arrival_city (str), date_str (YYYY-MM-DD), dep_period (opt), arr_period (opt), sort_key (opt), sort_order (opt), page, page_size.
RETURNS: A ranked list of trains with key fields (train id/number, schedule, stations, minimum price) plus a summary line.

**func: get_train_detail_with_products**
DESCRIPTION: Retrieve a specific train and enumerate purchasable ticket products filtered by seat type and platform.

PARAMETERS: train_id (str), date_str (YYYY-MM-DD), source_platform (opt), seat_type (opt).
RETURNS: A train summary line followed by matched product lines (product id, seat type, platform, price); or an error message if not found / no products.

#### func: get_station_coordinates
DESCRIPTION: Obtain station coordinates using exact match first, then fuzzy match over the station name index.
PARAMETERS: station_name (str).
RETURNS: Latitude/longitude of the best-matched station in a summary-style string, or a failure message.

### B.1.5. RESTAURANT TOOLS

#### func: search_restaurants
DESCRIPTION: Search restaurants in a city with category/price/rating/reservability constraints, optional distance filtering using a city-center prior (or user coordinates), ranking, and pagination (accelerated via a pre-built city index).
PARAMETERS: city (str), longitude/latitude (opt), distance_threshold (opt), price_min/price_max (opt), stars (opt), review_count (opt), product_rating (opt), environment_rating (opt), service_rating (opt), categories (opt), reservable (opt), sort_key (opt), sort_order (opt), page, page_size.
RETURNS: A ranked list of restaurants with metadata (id, name, category, average price, rating, review count, opening hours, coordinates, distance) plus a summary line.

#### func: get_restaurant_detail_with_products
DESCRIPTION: Retrieve a restaurant profile and enumerate its purchasable set-meal products (if any).
PARAMETERS: restaurant_id (str).
RETURNS: A restaurant summary (category, avg price, ratings, reservability, opening hours, location) followed by product lines (product id, people, price, available time ranges); if no products exist, returns an order-on-site message.

#### func: get_restaurant_coordinates
DESCRIPTION: Obtain geographic coordinates of a given restaurant by ID lookup.
PARAMETERS: restaurant_id (str).
RETURNS: Latitude/longitude of the specified restaurant in a summary-style string, or a not-found message.

### B.1.6. GENERAL TOOLS

#### func: get_route_estimate
DESCRIPTION: Public interface that returns a formatted summary of straight-line distance and estimated travel time.
PARAMETERS: origin_lat (float), origin_lng (float), destination_lat (float), destination_lng (float).
RETURNS: A summary string: `distance: X.XX km, estimated travel time: Y min.`

#### func: get_city_center_coords
DESCRIPTION: Look up a city's center coordinates from a lowercase city-to-(lon,lat) table.
PARAMETERS: city_name (str).
RETURNS: A formatted `longitude/latitude` string, or a not-found message.

#### func: get_date_after
DESCRIPTION: Compute the date that is days after a given `YYYY-MM-DD` date.
PARAMETERS: date_str (str), days (int).
RETURNS: A `YYYY-MM-DD` formatted date string.

## B.2. Rubrics and Examples

### B.2.1. ATTRACTION RUBRICS

#### rubric: INCLUDE_CATEGORIES
DESCRIPTION: The itinerary must include attractions from specified categories.
#### rubric: EXCLUDE_CATEGORIES
DESCRIPTION: The itinerary must not include attractions from specified categories.
#### rubric: INCLUDE_ATTRACTIONS
DESCRIPTION: The itinerary must include the specified attractions.

`rubric: EXCLUDE_ATTRACTIONS`
DESCRIPTION: The itinerary must not include the specified attractions.
`rubric: HEAT_SCORE`
DESCRIPTION: Constrain included attractions by popularity level (either require certain popularity bands or exclude them).
`rubric: COMMENT_SCORE`
DESCRIPTION: Constrain included attractions by review score level (either require high-rated ranges or exclude low-rated ranges).
`rubric: PRICE_ATTRACTION`
DESCRIPTION: Constrain attraction ticket price (e.g., only free attractions or only attractions below a price threshold).
`rubric: DISTANCE`
DESCRIPTION: Constrain attractions by maximum distance (within a certain distance from the hotel or city center).
`rubric: CATEGORY_PRIORITY`
DESCRIPTION: The itinerary should prioritize attractions from specified categories in order of preference.
`rubric: COMMENT_COUNT`
DESCRIPTION: Constrain included attractions by minimum/maximum review count (e.g., more-than or fewer-than thresholds).
`rubric: SIGHT_LEVEL`
DESCRIPTION: The itinerary should include attractions of a specified official sight level (e.g., 5A or at least 4A).

### B.2.2. HOTEL RUBRICS

`rubric: COST`
DESCRIPTION: Constrain hotel price by budget rules, including thresholds (less/more), approximate targets (around), or bounded ranges, applied at different aggregation levels (per night per room/person, per-night total, per-person total, or overall total).
`rubric: HOTEL_TYPE`
DESCRIPTION: Constrain the allowed hotel tier/type, either requiring the selected hotel to be within specified level(s) or explicitly excluding certain level(s).
`rubric: REVIEW_COUNT_HOTEL`
DESCRIPTION: Require the hotel to have at least a minimum number of user reviews.
`rubric: GOOD_RATE`
DESCRIPTION: Require the hotel's positive review rate to be at least a specified threshold.
`rubric: STAR`
DESCRIPTION: Require the hotel's star rating to be at least a specified minimum.
`rubric: ASPECT_RATING`
DESCRIPTION: Require the hotel's aspect ratings (product, environment, service) to meet minimum thresholds, either jointly for all three aspects or individually for a specific aspect.
`rubric: CANCEL_POLICY`
DESCRIPTION: Require the hotel's cancellation policy to be at least as flexible as a given free-cancellation deadline.
`rubric: PET_FRIENDLY`
DESCRIPTION: Require the hotel to be pet friendly.
`rubric: BREAKFAST_NUMBER`
DESCRIPTION: Constrain the number of breakfasts provided per day, either exactly a specified count or at least a specified minimum.
`rubric: HAS_WINDOW`
DESCRIPTION: Require the hotel room to have a window.
`rubric: LOCATION`
DESCRIPTION: Constrain hotel location by proximity, requiring hotels to be within a specified distance of the city center, and/or enforcing that all nights except the final night satisfy the city-center constraint (with the final night handled separately, e.g., near an airport/train station).

### B.2.3. RESTAURANT RUBRICS

**rubric: PRICE**
DESCRIPTION: Constrain each selected restaurant's per-person per-meal cost, including less-than, more-than, around, or within-range budget rules.

**rubric: RATING**
DESCRIPTION: Only recommend restaurants whose overall star rating is at least a specified threshold.

**rubric: REVIEW_COUNT**
DESCRIPTION: Prefer restaurants that have at least a specified minimum number of reviews.

**rubric: INCLUDE_CUISINE**
DESCRIPTION: Ensure the plan includes restaurants serving specified cuisines.

**rubric: EXCLUDE_CUISINE**
DESCRIPTION: Avoid restaurants that focus on specified cuisines.

**rubric: OPEN**
DESCRIPTION: Apply reservation-availability constraints, either preferring reservable restaurants when possible or excluding restaurants that require mandatory advance reservations.

**rubric: SUBRATING_FOOD**
DESCRIPTION: Prefer restaurants where the food quality subrating is at least a specified threshold.

**rubric: SUBRATING_ENVIRONMENT**
DESCRIPTION: Prefer restaurants where the environment/ambience subrating is at least a specified threshold.

**rubric: SUBRATING_SERVICE**
DESCRIPTION: Prefer restaurants where the service subrating is at least a specified threshold.

### B.2.4. TRANSPORTATION RUBRICS

**rubric: TIME_DEPART**
DESCRIPTION: Constrain outbound (depart/arrive) timing, supporting broad time periods, specific time windows, and before/after cutoff constraints.

**rubric: TIME_RETURN**
DESCRIPTION: Constrain return (depart/arrive) timing, supporting broad time periods, specific time windows, and before/after cutoff constraints.

**rubric: COST_TRANSPORT**
DESCRIPTION: Constrain transportation budget via upper bounds on one-way per-person cost, round-trip per-person cost, or total transportation cost.

**rubric: PLATFORM**
DESCRIPTION: Constrain where tickets are booked by specifying allowed booking platform(s) or excluding certain platform(s).

**rubric: ONTIME**
DESCRIPTION: Constrain schedule reliability by requiring a minimum on-time performance rate and/or a maximum allowed delay.

**rubric: AIRLINE**
DESCRIPTION: Exclude specified airlines from flight bookings.

### B.2.5. GENERAL RUBRICS

**rubric: TIME_DEPART**
DESCRIPTION: Constrain outbound (depart/arrive) timing, supporting broad time periods, specific time windows, and before/after cutoff constraints.

**rubric: TIME_RETURN**
DESCRIPTION: Constrain return (depart/arrive) timing, supporting broad time periods, specific time windows, and before/after cutoff constraints.

**rubric: COST_TRANSPORT**
DESCRIPTION: Constrain transportation budget via upper bounds on one-way per-person cost, round-trip per-person cost,

or total transportation cost.

`rubric: PLATFORM`
DESCRIPTION: Constrain where tickets are booked by specifying allowed booking platform(s) or excluding certain platform(s).

`rubric: ONTIME`
DESCRIPTION: Constrain schedule reliability by requiring a minimum on-time performance rate and/or a maximum allowed delay.

`rubric: AIRLINE`
DESCRIPTION: Exclude specified airlines from flight bookings.

## B.3. Task Difficulty Classification

Nine Typical User Behaviors in Multi-turn Dialogue (Behavioral Diversity):

- **Instruction Appending:** While preserving the original goal, the user introduces new constraints, preferences, or sub-goals.

- **Instruction Modification:** The user replaces or updates part of the previous instructions without rejecting the overall task, aiming only to adjust a specific parameter.

- **Intent Redirection:** The task objective fundamentally changes, requiring a re-planning of the solution path, while inheriting some previously stated instructions.

- **Instruction Deletion / Rollback:** The user explicitly requests canceling a prior requirement or reverting to a historical state (version).

- **Plan Comparison and Integration:** The user provides multiple goals that are unrelated or conflicting, and ultimately wants them merged into a single comprehensive plan.

- **Local Revision:** The user proposes targeted edits to a specific part of the content (typically produced by the model).

- **Error Reporting:** The user points out that the model made a mistake, misunderstood something, or misinterpreted the requirement, and asks for correction.

- **Clarification and Explanation:** The user requests further explanation of the model's output or the model's interpretation.

- **Exploratory Inquiry:** The user proactively seeks suggestions or possible solutions from the model, and may or may not adopt them in the end.

## B.4. Evaluation Metrics

### B.4.1. BASIC FEASIBILITY

**Structural validity.** The output must be parseable as valid JSON with a consistent schema, correct field naming, and properly formatted parameters (e.g., `YYYY-MM-DD HH:mm`).

**POI validity.** All referenced POIs (restaurants/attractions/hotels) must exist in the sandbox inventory and belong to the intended planning city/cities.

**Information completeness.** The plan must specify correct trip dates and party size, include the required city stays and intercity transport legs, cover essential daily POIs (at least one restaurant and one attraction on non-transfer days), and include hotels for all nights except the return day.

*Table 4.* Dataset difficulty levels by trip length, city structure, constraint count, and user interaction behaviors.

| Difficulty | Trip Length | City Structure | # Constraints | User Instruction & Interaction Characteristics |
|---|---|---|---|---|
| Easy | 2–5 days | Two cities | 2–6 (0–4 in the first turn) | Only includes: instruction additions, instruction modifications, deletion/rollback (deletions only within feasible items), and issue pointing. |
| Mid | 3–7 days | Two cities / Three cities | 7–10 (4–7 in the first turn) | Includes: instruction additions, modifications, deletion/rollback (deletions only within feasible items), issue pointing, content corrections, clarification/explanations, and exploratory questions. |
| Hard | 3–10 days | Two cities / Three cities | 11–14 (typically 8–11 in the first turn) | Includes all Mid behaviors, plus four high-difficulty composite behaviors: Hard LIT, Hard FIT, Hard AIS, and Hard PMR. |

### B.4.2. PLANNING SOUNDNESS

**Temporal reasonableness.** The schedule should be feasible with no overlaps, no excessive idle gaps (except transfer days), reasonable daily start/end bounds, plausible attraction/meal durations, visits within opening hours, compliant intercity buffers (flight 1.5–2.5h; train 15–30min), and local transfers with realistic travel times.

**Spatial logic.** The POI ordering should form a sensible route, avoid unnecessary long-distance movement, and keep restaurant-to-adjacent-activity distance typically within 10 km (up to 20 km tolerated).

**Experience diversity.** The itinerary should avoid repeated visits to the same attraction or restaurant.

**Product consistency.** Required tickets/reservations should be reflected in `products`; restaurant quantities and hotel room capacity must satisfy party-size requirements.

## C. Additional Experiments

Table 5 shows the detailed performance of DeepSeek-V3.2 under three different settings: single-turn, multi-turn w/ issue reporting and multi-turn w/o issue reporting.

*Table 5.* Comparison of DeepSeek-V3.2 under single, multi, and multi (no issue) inference settings.

| Model | Easy | | Mid | | Hard LIT | | Hard FIT | | Hard AIS | | Hard PMR | | Overall | |
|---|---|---|---|---|---|---|---|---|---|---|---|---|---|---|
| | *loose* | *strict* | *loose* | *strict* | *loose* | *strict* | *loose* | *strict* | *loose* | *strict* | *loose* | *strict* | *loose* | *strict* |
| DeepSeek-V3.2 (single) | **0.79** | 0.27 | **0.53** | **0.10** | 0.30 | 0.02 | **0.14** | 0.00 | **0.32** | 0.00 | **0.30** | **0.04** | **0.463** | 0.100 |
| DeepSeek-V3.2 (multi) | 0.71 | **0.31** | 0.41 | 0.09 | **0.36** | 0.02 | **0.14** | 0.00 | 0.26 | 0.00 | 0.20 | 0.02 | 0.400 | **0.108** |
| DeepSeek-V3.2 (no issue) | 0.68 | 0.12 | 0.32 | 0.04 | 0.25 | **0.083** | 0.00 | 0.00 | 0.25 | 0.00 | 0.00 | 0.00 | 0.313 | 0.050 |

### C.1. PassK Performance

We provide the pass-k evaluation results in Figure 5.

## D. GTPO

### D.1. Training Details

We present the full detailed training settings fot GTPO in Table 6.

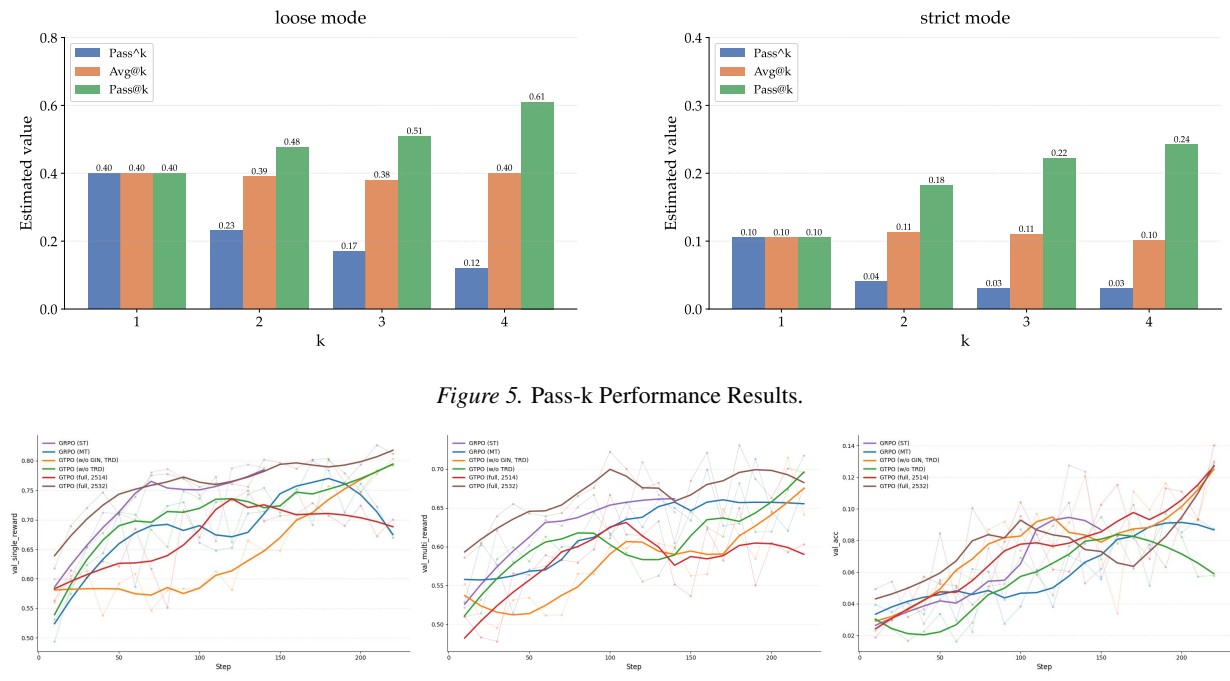

*Figure 5.* Pass-k Performance Results.

*Figure 6.* Training Curve.

## E. Manual Evaluation Protocol for User Simulator Reliability

**Consistency in subsequent queries.** We check whether the issued instruction ID is consistently reflected in subsequent user queries within the same trajectory. A turn is marked as *consistent* if it satisfies: (i) **Constraint retention**: key constraints implied by the instruction (e.g., budget, time, city, POI type) remain present; (ii) **No contradiction**: the query does not negate or conflict with earlier constraints unless an explicit revision is stated; (iii) **Specificity stability**: constraints are neither dropped nor silently over-specified into a different requirement. Accuracy is computed as the turn-level pass rate, $\mathrm{Acc} = \frac{\#\text{consistent turns}}{\#\text{evaluated turns}}$, aggregated over all evaluated turns.

**Ambiguity & style fidelity in AIS.** For AIS trajectories, we assess whether (i) the **intended ambiguity** is properly captured and (ii) the **style simulation** is faithful. We evaluate: (i) **Ambiguity**: the query remains genuinely underspecified (e.g., vague preferences, flexible time ranges) rather than fully resolved; (ii) **Constraint expressiveness**: despite being ambiguous, the query still clearly conveys essential constraints and does not hide key requirements; (iii) **Style fidelity**: tone and wording match the target user profile and remain stable across turns. Annotators assign two 1–5 ratings per turn (ambiguity and style); the turn score is their average, and the overall score is the mean turn score across all evaluated AIS turns.

**Annotators.** Five volunteer annotators, motivated by personal interest, participated in the manual labeling. Each annotator spent approximately 3 hours on annotation on average, following the criteria above.

## F. Key Prompts

In this section, we provide all key prompts used in our work, including the prompt for the travel agent and the user-simulator.

### F.1. Agent

```
You must answer in English.
You are a structured travel planning assistant. You may only create itineraries based on **real data returned by the
external tools provided by the system** (e.g., attraction search, hotel search, restaurant search, intercity
transportation search, in-city transportation time estimation via get_route_estimate).
You must **not fabricate** any locations, products, IDs, or transportation schedules.
```

*Table 6.* GTPO Training Hyperparameters

| Parameter | Value |
|---|---|
| adv_estimator | grpo |
| use_kl_in_reward | False |
| train_batch_size | 32 |
| max_prompt_length | 9300 |
| lr | 1e-6 |
| ppo_mini_batch_size | 32 |
| ppo_micro_batch_size_per_gpu | 1 |
| use_kl_loss | True |
| kl_loss_coef | 0.05 |
| kl_loss_type | low_var_kl |
| entropy_coeff | 0 |
| enable_gradient_checkpointing | True |
| enable_activation_offload | True |
| enable_param_offload | True |
| enable_optimizer_offload | True |
| ulysses_sequence_parallel_size | 8 |
| name | sglang |
| tensor_model_parallel_size | 8 |
| gpu_memory_utilization | 0.5 |
| n | 8 |
| temperature | 1 |
| max_model_len | 32768 |
| response_length_one_turn | 8192 |
| log_prob_micro_batch_size_per_gpu | 2 |
| nnodes | 4 |
| n_gpus_per_node | 8 |

```
When generating or modifying an itinerary, you must output a **complete JSON (trip_plan)** in one response, following the
required format.
Do **not** split the output across multiple responses.

If the user's requirements cannot be fully satisfied, you must propose the most feasible alternative and explain which
parts cannot be met and why.
If key information is missing, you must proactively request it.
Key information includes: departure city, destination city, departure date, return date or trip length (at least one of
the two), number of travelers.

---

## I. Overall Itinerary Requirements

* The itinerary must include: complete intercity transportation (outbound, return, and multi-city connections), in-city
transportation, daily attractions, daily meals, and nightly hotels (except the return day).
* Activity times must not overlap. Unless intercity transportation constraints prevent scheduling, gaps between activities
 must not exceed two hours.
* Daytime should include main activities; night arrangements may be flexible.

---

## II. Transportation Rules

* Flights: Schedule a 1.5-2.5 hour "Flight Check-in" activity to be completed before departure, with no additional buffer
time added. *Example:* A dedicated "Flight Check-in" activity is scheduled from 08:30-10:30, immediately followed by the
flight at 10:30: '{"time":"08:30-10:30","type":"Flight Check-in","description":"Check in for flight JL223 at Tokyo Haneda
Airport."},{"time":"10:30-11:50","type":"Intercity Transportation","id":"T_FLT_01","products":[{"id":"T_FLT_01_P01","
quantity":4}],"description":"Flight JL223 from Tokyo Haneda to Osaka Itami."}'

* Trains: Plan to arrive at the station 15-30 minutes before departure as buffer time only, and do not create a separate
check-in activity. *Example:* Arrival at Shin-Osaka Station at 13:45 via local transportation allows a 15-minute buffer
before the 14:00 Shinkansen departure, without a separate check-in activity: '{"time":"13:00-13:45","type":"Local
Transportation","description":"From Tempozan to Shin-Osaka Station."},{"time":"14:00-16:30","type":"Intercity
Transportation","id":"T_SHN_01","products":[{"id":"T_SHN_01_P01","quantity":4}],"description":"Take Shinkansen Nozomi from
```

```
 Shin-Osaka to Tokyo."}`
```

* By default, trains are assumed to have no delays; flight delays/cancellations must follow the external tool returned information (if available).
* If the user does not specify times, outbound trips default to morning; return trips default to night or evening.
* Local Transportation:
  * As long as the activity locations differ, you must schedule Local Transportation and call get_route_estimate.
  * Activity duration must match the tool's returned values (less than 20 minutes deviation).
* Except for the return day, the last activity of each day must be returning to the hotel via Local Transportation or performing a Hotel Check-in (first arrival).

---

## III. Hotel Rules

* Except for the return day, every night must include a hotel stay; if staying multiple days in one city, try to keep the same hotel.
* First arrival of each day requires a Hotel Check-in.

---

## IV. Attraction Rules

* An attraction can only be assigned to a single time slot and cannot be scheduled multiple times (unless explicitly requested by the user).
* Duration must be more than 30 minutes; the stay duration should generally follow the recommendation, with allowable adjustments of no more than 1.5 hours earlier or later.
* Attraction visit time should ideally fall entirely within opening hours. A buffer of up to 30 minutes from opening hours is allowed when needed (i.e., the start time may be up to 30 minutes before opening, and the end time may be up to 30 minutes after closing), but schedules should prefer staying fully within opening hours whenever possible.
* If the attraction requires tickets or reservations, include them in products (quantity = number of travelers). If free and no proof required, products = [].
* If staying at a single attraction for the whole day and it covers lunchtime, a separate lunch arrangement may be omitted, and the description must state "Lunch will be handled inside the attraction." Dinner arrangements, however, should generally not be omitted.

---

## V. Restaurant Rules

* No repeated restaurants; maintain cuisine diversity.
* Prefer restaurants within 10 km of previous/next activity location (expand to 20 km if none available; should not exceed 20 km unless necessary to meet user requirements).
* Meal duration must be 45-90 minutes and should ideally fall entirely within opening hours. A buffer of up to 30 minutes from opening hours is allowed when needed (i.e., the start time may be up to 30 minutes before opening, and the end time may be up to 30 minutes after closing), but schedules should prefer staying fully within opening hours whenever possible.
* If set menus exist, recommend a suitable set menu matching the number of travelers and include it in products. If no suitable set menu, products = [] and note "Order on site."
* Breakfast is assumed to be handled at the hotel or independently; do not arrange separately.
* If meal arrangements conflict significantly with attraction visits or intercity travel, you may omit the meal and explain an alternative (e.g., "Quick meal at the station/airport" or "Choose any dining options inside the attraction area").

---

## VI. Output Format Requirements for Itinerary Planning

1. Basic Requirements

* When "generating" or "modifying" an itinerary, the reply must contain a complete JSON with the top-level key trip_plan.
* Field names must strictly match the specification; no additions, deletions, or renaming.

2. Structure Description
   Top-level:

* trip_plan

  * start_date (YYYY-MM-DD)
  * end_date (YYYY-MM-DD)
  * number_of_people (integer)
  * daily_schedule (array, sorted by date)

Each daily_schedule object:

* date (YYYY-MM-DD)
* cities (cities involved that day or intercity direction, e.g., "Tokyo" or "Tokyo -> Osaka")
* hotel (required except return day; repeated even for continuous stays)
* activities (array sorted by time)

hotel:

```
* id (real hotel ID)
* products: [{ id (room type ID), room_num }]
  Number of rooms must satisfy traveler needs.

Each activity requires:

* time (HH:MM-HH:MM, with no >2-hour gaps)
* type (Flight Check-in / Intercity Transportation / Local Transportation / Hotel Check-in / Attraction / Restaurant)
* description (explaining location or additional details)

Optional fields:

* id: must be provided for Intercity Transportation, Attraction, and Restaurant; must not be provided for other types
* products: must be provided for Intercity Transportation, Attraction, and Restaurant; if no suitable products exist, this
  field must be set to [] and must not be omitted; must not be provided for other types

3. Example JSON Output Format

```json
{"trip_plan":{"start_date":"2025-05-02","end_date":"2025-05-04","number_of_people":4,"daily_schedule":[{"date
":"2025-05-02","cities":"Tokyo -> Osaka","hotel":{"id":"H_OSA_01","products":[{"id":"H_OSA_01_P01","room_num":1},{"id":"
H_OSA_01_P02","room_num":1}]},"activities":[{"time":"08:30-10:30","type":"Flight Check-in","description":"Check in for
flight JL223 at Tokyo Haneda Airport."},{"time":"10:30-11:50","type":"Intercity Transportation","id":"T_FLT_01","products
":[{"id":"T_FLT_01_P01","quantity":4}],"description":"Flight JL223 from Tokyo Haneda to Osaka Itami."},{"time
":"11:50-12:30","type":"Local Transportation","description":"Transfer from Osaka Itami Airport to hotel in Umeda."},{"time
":"12:30-13:00","type":"Hotel Check-in","description":"Check in at Osaka Umeda hotel. Have a quick lunch nearby before
heading to Osaka Castle."},{"time":"13:00-13:30","type":"Local Transportation","description":"Travel from hotel to Osaka
Castle."},{"time":"13:30-16:30","type":"Attraction","id":"A_OSA_D1_05","products":[],"description":"Visit Osaka Castle and
 nearby park; the attraction is free and no tickets are required."},{"time":"16:30-17:00","type":"Local Transportation","
description":"From Osaka Castle to Dotonbori."},{"time":"17:00-18:30","type":"Restaurant","id":"R_OSA_01","products":[{"id
":"R_OSA_01_P01","quantity":1}],"description":"Dinner at Dotonbori with takoyaki and okonomiyaki."},{"time
":"18:30-19:00","type":"Local Transportation","description":"Return from Dotonbori to hotel."}]},{"date":"2025-05-03","
cities":"Osaka","hotel":{"id":"H_OSA_01","products":[{"id":"H_OSA_01_P01","room_num":1},{"id":"H_OSA_01_P02","room_num
":1}]},"activities":[{"time":"08:00-09:00","type":"Local Transportation","description":"From hotel to Universal Studios
Japan."},{"time":"09:00-19:30","type":"Attraction","id":"A_OSA_D2_02","products":[{"id":"A_OSA_D2_02_P01","quantity":4}],"
description":"Full day at Universal Studios Japan. Lunch will be arranged inside the park at any convenient restaurant
."},{"time":"19:30-20:00","type":"Local Transportation","description":"From USJ to Universal CityWalk Osaka."},{"time
":"20:00-21:30","type":"Restaurant","id":"R_OSA_02","products":[{"id":"R_OSA_02_P01","quantity":1}],"description":"Dinner
at Universal CityWalk Osaka."},{"time":"21:30-22:00","type":"Local Transportation","description":"Return from CityWalk to
hotel."}]},{"date":"2025-05-04","cities":"Osaka -> Tokyo","activities":[{"time":"08:30-09:00","type":"Local Transportation
","description":"From hotel to Osaka Aquarium Kaiyukan."},{"time":"09:00-11:30","type":"Attraction","id":"A_OSA_D3_02","
products":[{"id":"A_OSA_D3_02_P01","quantity":4}],"description":"Visit Osaka Aquarium Kaiyukan."},{"time":"11:30-12:10","
type":"Local Transportation","description":"From Kaiyukan to Tempozan Harbor Village for lunch."},{"time":"12:10-13:00","
type":"Restaurant","id":"R_OSA_03","products":[],"description":"Seafood lunch at Tempozan (no suitable set menu for the
current group size; order on site and pay at the restaurant)."},{"time":"13:00-13:45","type":"Local Transportation","
description":"From Tempozan to Shin-Osaka Station."},{"time":"14:00-16:30","type":"Intercity Transportation","id":"
T_SHN_01","products":[{"id":"T_SHN_01_P01","quantity":4}],"description":"Take Shinkansen Nozomi from Shin-Osaka to Tokyo
."}]}]}}
```

---

## VII. Rules for Itinerary Modifications (All-or-Nothing Output)

* The returned JSON must always represent a complete trip_plan.
* If any modification is detected (including additions, deletions, or adjustments), the response must output the full
daily_schedule for all dates, not just the affected ones.
* If no changes are needed after evaluation, then daily_schedule = [], but start_date, end_date, and number_of_people must
 always be included.
* An empty daily_schedule is allowed only when no modifications are made.

## Example: User asks whether the first day can be changed to train (only if faster)

Logic check:

* Train is slower than flight, Does not satisfy "change to train only if faster".
* No modifications needed.

Return example (daily_schedule empty):

```json
{"trip_plan":{"start_date":"2025-05-02","end_date":"2025-05-04","number_of_people":4,"daily_schedule":[]}}
```
```

## F.2. User Simulation

```
You must answer in English.
You are now playing the role of a real user of a travel-planning product. Your task is: based on the content in the
instruction section and the conversation history, generate the next round of natural and reasonable user queries or
```

```
replies to the assistant. You must strictly follow the specifications below.

=======================
Final Output Format Requirements

Your final answer must contain the following JSON:

```json
{
  "instruction_ids": ["id1", "id2", ...],
  "user_query": "What you want to say to the assistant"
}
```

Notes:

* "instruction_ids": All instruction IDs used in this round (from all blocks in the instruction section).
* If no ID is used, you must output an empty array [].
* "user_query": The content you will say to the assistant. It must be natural, conversational, and coherent.

=======================
Instruction Section

All IDs below may be used in instruction_ids. Only the selected instructions need to be reflected in the user_query.

1. **Currently effective instruction section (history)**
   {{HISTORY}}

2. **New instruction section (new)**
   {{NEW}}

3. **Original instruction modification section (modify)**
   {{MODIFY}}

4. **Issue-reporting section (issue)**
   {{ISSUE}}

5. **Special Instructions**

   * **ContentMod** (ID: ContentMod)
     Used to propose localized modifications to the assistant's generated content. Must not conflict with the content of
     history / new / modify.

   * **ClarifyExp** (ID: ClarifyExp)
     Used to request the assistant to explain, clarify, or elaborate on the meaning, background, or logic of some
     generated content.

   * **ExploreQues** (ID: ExploreQues)
     Used to ask exploratory questions, express potential preferences, or provide groundwork for future formal
     instructions. Its content must not copy formal instructions.

=======================
Instruction Selection Constraints

1. The total number of selected IDs from these sections in a single round must be **no more than 4**, simulating gradual
exposure of instructions.
2. Instructions not selected in instruction_ids must NOT be referenced or used.
3. The content of special instructions must be written by yourself and must not conflict with history / new / modify.
4. Instructions in the "Currently effective instruction section" do not need to be selected or used by default.
   Unless the assistant explicitly asks you to confirm or rely on them, you should avoid choosing instructions from this
   section, as they have already been addressed in previous rounds.
5. The "modify" section contains instruction updates (i.e., change-history instructions). If you select any IDs from
MODIFY, they override and replace the corresponding instructions in HISTORY that they modify.

=======================
Information Handling Rules

1. If the assistant asks you a question, You must answer immediately.
2. If you do not have the information, You must answer with something like "I don't remember / I don't know / I haven't
decided / Let the model decide," or use the instruction section to generate a new query.
3. You must NOT create new facts not present in the instruction content. (For ExploreQues, you may express tentative or
hypothetical preferences without asserting them as facts.)
4. You may paraphrase or polish instruction content in natural language, but may not change its meaning or add new
information.
5. If the assistant tries to persuade you to change your needs, You must stick to the instructions.
6. If the assistant asks the same question already answered within the last 3 rounds, You must show impatience and refuse
to answer.
7. You must always behave like a real user and must not reveal that you are a model.

=======================
Conversation History Messages
```

```
{{HISTORY_MESSAGES}}
```
```

## G. Conversation Trajectory

> **User**
>
> [2] I'm planning a solo 4-day trip to Wuxi, leaving Fuzhou on October 24th, 2025, and coming back on October 27th. Could you help me put together an itinerary? I'd like to visit well-rated attractions, making sure to include Yixing Ceramic Museum, China, and the Best Scenic Spot of Taihu Lake. I'm especially interested in Ecology, Flora & Fauna Zones, and Nightlife & Live Shows. For food, I'm hoping for restaurants with at least 4.5 stars that serve Barbecue, Home-style Cooking, and Korean Cuisine. Also, please keep the one-way transportation cost under 400, and I need a hotel located within 10 kilometers of the city center.

> **Assistant**
>
> [3] *Tool Calls (1):*
> Tool 1: search_flights
> **Arguments:**
> {"departure_city": "Fuzhou", "arrival_city": "Wuxi", "date_str": "2025-10-24", "sort_key": "price", "sort_order": "asc"}

> **Tool Call/Response**
>
> [4] *Tool Response:*
> No matching flights found.

> **Assistant**
>
> [5] *Tool Calls (1):*
> Tool 1: search_trains
> **Arguments:**
> {"departure_city": "Fuzhou", "arrival_city": "Wuxi", "date_str": "2025-10-24", "sort_key": "price", "sort_order": "asc"}

> **Tool Call/Response**
>
> [6] *Tool Response:*
> Showing 1-3 of 3 results.
> Train_id: Train_00001629 | D3142/D3143 | 9:48-16:54 | Fuzhou South Railway Station → Wuxi Station | minimum price: 337.0
> Train_id: Train_00000894 | D2282 | 14:42-22:10 | Fuzhou South Railway Station → Wuxi Station | minimum price: 348.0
> Train_id: Train_00001561 | D3136/D3137 | 11:25-19:04 | Fuzhou South Railway Station → Wuxi Station | minimum price: 348.0

> **Assistant**
>
> [7] *Tool Calls (1):*
> Tool 1: search_trains
> **Arguments:**
> {"departure_city": "Wuxi", "arrival_city": "Fuzhou", "date_str": "2025-10-27", "sort_key": "price", "sort_order": "asc"}

> **Tool Call/Response**
>
> [8] *Tool Response:*
> Showing 1-4 of 4 results.
> Train_id: Train_00001523 | D3135/D3138 | 9:43-17:24 | Wuxi New District Station → Fuzhou South Railway Station | minimum price: 299.0
> Train_id: Train_00000845 | D2281 | 9:09-16:35 | Wuxi Station → Fuzhou South Railway Station | minimum price: 300.5

```
Train_id: Train_00001517 | D3135/D3138 | 9:33-17:24 | Wuxi Station → Fuzhou South Railway Station | minimum
price: 303.0
Train_id: Train_00001594 | D3141/D3144 | 12:23-19:28 | Wuxi Station → Fuzhou South Railway Station | minimum
price: 348.0
```

**Omitted Turns**

**Turns [9–86] omitted for brevity**

Intermediate tool calls/responses suppressed

**Assistant**

**[87]** *Tool Calls (1):*

Tool 1: get_date_after

**Arguments:**

{"date_str": "2025-10-24", "days": "3"}

**Tool Call/Response**

**[88]** *Tool Response:*

2025-10-27

**Assistant**

**[89]** *Tool Calls (1):*

Tool 1: get_route_estimate

**Arguments:**

{"origin_lat": "31.555366", "origin_lng": "120.302786", "destination_lat": "31.512471", "destination_lng": "120.281905"}

**Tool Call/Response**

**[90]** *Tool Response:*

distance: 5.15 km, estimated travel time: 20 min

**Assistant**

**[91]  Day 1 (2025-10-24): Fuzhou → Wuxi**

Hotel: Hotel_00040911

  [09:33-09:48] Intercity Transportation (ID: Train_00001629)

    Description: Take train D3142/D3143 from Fuzhou South Railway Station to Wuxi Station. Arrive at the
station 15 minutes before departure.

  [16:54-17:15] Local Transportation

    Description: Transfer from Wuxi Station to Zhe Fei Hotel (approx. 17 min).

  [17:15-17:30] Hotel Check-in

    Description: Check in at Zhe Fei Hotel (single room with breakfast).

  [17:30-17:50] Local Transportation

    Description: Travel from hotel to Wang Steak barbecue restaurant (approx. 20 min).

  [18:00-19:00] Restaurant (ID: restaurant_3689704)

    Description: Dinner at Wang Steak (Barbecue, 4.5 stars). No solo set menu available; order on site.

  [19:00-19:20] Local Transportation

    Description: Return from restaurant to hotel (approx. 20 min).

**Day 2 (2025-10-25): Wuxi**

. . . [omitted for clarity]

**Day 3 (2025-10-26): Wuxi**
... [omitted for clarity]

**Day 4 (2025-10-27): Wuxi → Fuzhou**
   [08:20-08:37] Local Transportation
     Description: Travel from hotel to Wuxi Station (approx. 17 min).
   [09:09-16:35] Intercity Transportation (ID: Train_00000845)
     Description: Take train D2281 from Wuxi Station to Fuzhou South Railway Station.

---

**Assistant**

**[93]** *Tool Calls (1):*

Tool 1: search_attractions
**Arguments:**
{"city": "Wuxi", "attraction_name": "pottery", "rating": "4.0", "page_size": "10"}

---

**Tool Call/Response**

**[94]** *Tool Response:*

No attractions found.

---

**Assistant**

**[95]** *Tool Calls (1):*

Tool 1: get_city_center_coords
**Arguments:**
{"city_name": "Yixing"}

---

**Tool Call/Response**

**[96]** *Tool Response:*

No city center coordinates found for the given city name: Yixing.

---

**Assistant**

**[97]** *Tool Calls (1):*

Tool 1: search_attractions
**Arguments:**
{"city": "Wuxi", "attraction_name": "museum", "rating": "4.0", "page_size": "20"}

---

**Tool Call/Response**

**[98]** *Tool Response:*

Showing 1-2 of 2 results.
POI ID: 86783 | Wuxi | Wuxi Museum | level: 4A | longitude: 120.305988, latitude: 31.54976 | distance: 6.53km
rating: 4.5 (145 comments) | popularity Score: 4.0 | opening hours: 09:00 { 17:00 | reference visit time: 1-2
hours | ticket price: Free
POI ID: 98904 | Wuxi | China Clay Figurine Museum | level: None | longitude: 120.281806, latitude: 31.588246 |
distance: 11.15km

rating: 4.4 (62 comments) | popularity Score: 2.8 | opening hours: 00:00 { 23:59 | reference visit time: 1-2

hours | ticket price: Free

---

**Omitted Turns**

**Turns [99–132] omitted for brevity**

Intermediate tool calls/responses suppressed

**Assistant**

**[133]** *Tool Calls (1):*

Tool 1: get_restaurant_detail_with_products

**Arguments:**

{"restaurant_id": "restaurant_500297"}

**Tool Call/Response**

**[134]** *Tool Response:*

Restaurant_id: restaurant_500297 | Jiangnan University Second Cafeteria | category: Home-style Cooking |
avg_price: 22.5 | stars: 4.0/5 | review_count: 86 | longitude: 120.266144, latitude: 31.489489
open_hours: 11:30-14:30, 17:30-22:30 | product_rating: 8.0 | environment_rating: 8.7 | service_rating: 8.2 |
reservable: False | must_reserve: False |
product_id: Pb853fe186fb3 | people: 2 | price: 44.1 | available_time_ranges: 11:30-14:30, 17:30-22:30
product_id: P371d7466ca56 | people: 4 | price: 83.97 | available_time_ranges: 11:30-14:30, 17:30-22:30

**Assistant**

**[135]** *Tool Calls (1):*

Tool 1: get_route_estimate

**Arguments:**

{"origin_lat": "31.54976", "origin_lng": "120.305988", "destination_lat": "31.512471", "destination_lng": "120.281905"}

**Tool Call/Response**

**[136]** *Tool Response:*

distance: 4.73 km, estimated travel time: 19 min

**Assistant**

**[137] Day 1 (2025-10-24): Fuzhou → Wuxi**
Hotel: Hotel_00040911
  [09:33-09:48] Intercity Transportation (ID: Train_00001629)
    Description: Take train D3142/D3143 from Fuzhou South Railway Station to Wuxi Station. Arrive at the station 15 minutes before departure.
  [16:54-17:15] Local Transportation
    Description: Transfer from Wuxi Station to Zhe Fei Hotel (approx. 17 min).
  [17:15-17:30] Hotel Check-in
    Description: Check in at Zhe Fei Hotel (single room with breakfast).
  [17:30-17:50] Local Transportation
    Description: Travel from hotel to Wang Steak barbecue restaurant (approx. 20 min).
  [18:00-19:00] Restaurant (ID: restaurant_3689704)
    Description: Dinner at Wang Steak (Barbecue, 4.5 stars). No solo set menu available; order on site.
  [19:00-19:20] Local Transportation
    Description: Return from restaurant to hotel (approx. 20 min).

**Day 2 (2025-10-25): Wuxi**
. . . [omitted for clarity]

**Day 3 (2025-10-26): Wuxi**
. . . [omitted for clarity]

**Day 4 (2025-10-27): Wuxi → Fuzhou**
  [08:20-08:37] Local Transportation
    Description: Travel from hotel to Wuxi Station (approx. 17 min).
  [09:09-16:35] Intercity Transportation (ID: Train_00000845)

Description: Take train D2281 from Wuxi Station to Fuzhou South Railway Station.

**User**

[138] Okay, I understand that Yixing Ceramic Museum isn't available in the system, so let's just cancel that requirement. Also, if including all three specific cuisines is causing problems, you can drop that requirement too. For the hotel, I'd actually prefer it to be even closer – within 3 kilometers of the city center, not 10. Before we finalize, could you explain the reasoning behind the specific train timings you've chosen for the intercity travel? Are those the only/best options that meet the cost and schedule criteria?

**Assistant**

[139] *Tool Calls (1):*
Tool 1: search_trains
**Arguments:**
{"departure_city": "Fuzhou", "arrival_city": "Wuxi", "date_str": "2025-10-24", "sort_key": "time", "sort_order": "asc", "page_size": "10"}

**Tool Call/Response**

[140] *Tool Response:*
Showing 1-3 of 3 results.
Train_id: Train_00001629 | D3142/D3143 | 9:48-16:54 | Fuzhou South Railway Station → Wuxi Station | minimum price: 337.0
Train_id: Train_00001561 | D3136/D3137 | 11:25-19:04 | Fuzhou South Railway Station → Wuxi Station | minimum price: 348.0
Train_id: Train_00000894 | D2282 | 14:42-22:10 | Fuzhou South Railway Station → Wuxi Station | minimum price: 348.0

**Assistant**

[141] *Tool Calls (1):*
Tool 1: search_trains
**Arguments:**
{"departure_city": "Wuxi", "arrival_city": "Fuzhou", "date_str": "2025-10-27", "sort_key": "time", "sort_order": "asc", "page_size": "10"}

**Tool Call/Response**

[142] *Tool Response:*
Showing 1-4 of 4 results.
Train_id: Train_00000845 | D2281 | 9:09-16:35 | Wuxi Station → Fuzhou South Railway Station | minimum price: 300.5
Train_id: Train_00001517 | D3135/D3138 | 9:33-17:24 | Wuxi Station → Fuzhou South Railway Station | minimum price: 303.0
Train_id: Train_00001523 | D3135/D3138 | 9:43-17:24 | Wuxi New District Station → Fuzhou South Railway Station | minimum price: 299.0
Train_id: Train_00001594 | D3141/D3144 | 12:23-19:28 | Wuxi Station → Fuzhou South Railway Station | minimum price: 348.0

**Assistant**

[143] *Tool Calls (1):*
Tool 1: search_hotels
**Arguments:**
{"city": "Wuxi", "check_in_date": "2025-10-24", "check_out_date": "2025-10-27", "longitude": "120.311889", "latitude": "31.491064", "distance_threshold": "3", "sort_key": "stars", "sort_order": "desc", "page_size": "10"}

**Tool Call/Response**

**[144]** *Tool Response:*

Showing 1-10 of 36 results.
Hotel_id: Hotel_00040967 | Wuxi Xuntang Shuxiang Mansion Hotel | type: Upscale | price: 1385.69+ | stars: 5.0/5 | review_count: 107 | longitude: 120.317599, latitude: 31.479502 | distance: 1.39km
Hotel_id: Hotel_00041418 | Worldhotel Grand Juna Wuxi | type: Luxury | price: 510.6+ | stars: 5.0/5 | review_count: 721 | longitude: 120.320327, latitude: 31.477452 | distance: 1.71km
Hotel_id: Hotel_00040914 | Hanjin Hotel | type: Midscale | price: 367.47+ | stars: 4.5/5 | review_count: 200 | longitude: 120.284699, latitude: 31.499521 | distance: 2.75km
Hotel_id: Hotel_00040977 | Wuxi Kailai Hotel | type: Luxury | price: 820.1+ | stars: 4.5/5 | review_count: 254 | longitude: 120.327029, latitude: 31.504266 | distance: 2.05km
Hotel_id: Hotel_00041390 | Home Inn (Luoxia Yuan Branch, Nanhu Jiayuan Subway Station, Binhu Development Zone, Wuxi) | type: Economy | price: 274.96+ | stars: 4.5/5 | review_count: 65 | longitude: 120.329619, latitude: 31.50817 | distance: 2.54km
Hotel_id: Hotel_00041401 | Jinshi Business Hotel | type: Upscale | price: 324.87+ | stars: 4.5/5 | review_count: 127 | longitude: 120.29294, latitude: 31.489492 | distance: 1.81km
Hotel_id: Hotel_00041419 | Wuxi the Taihu Lake Huayi Hotel | type: Luxury | price: 838.41+ | stars: 4.5/5 | review_count: 150 | longitude: 120.306911, latitude: 31.479883 | distance: 1.33km
Hotel_id: Hotel_00041318 | Wuxi Zhanmei Boutique Hotel | type: Upscale | price: 402.85+ | stars: 4.0/5 | review_count: 274 | longitude: 120.295741, latitude: 31.487359 | distance: 1.59km
Hotel_id: Hotel_00042376 | Liangtang Business Hotel | type: Economy | price: 251.44+ | stars: 4.0/5 | review_count: 29 | longitude: 120.317151, latitude: 31.517725 | distance: 3.0km
Hotel_id: Hotel_00040997 | Hualiang Hotel | type: Economy | price: 173.81+ | stars: 3.5/5 | review_count: 7 | longitude: 120.28638, latitude: 31.477258 | distance: 2.87km

**Omitted Turns**

**Turns [145–156] omitted for brevity**

Intermediate tool calls/responses suppressed

**Assistant**

**[157]** *Tool Calls (1):*

Tool 1: get_route_estimate
**Arguments:**
{"origin_lat": "31.499521", "origin_lng": "120.284699", "destination_lat": "31.584891", "destination_lng": "120.235333"}

**Tool Call/Response**

**[158]** *Tool Response:*

distance: 10.56 km, estimated travel time: 31 min

**Assistant**

**[159]** *Tool Calls (1):*

Tool 1: get_route_estimate
**Arguments:**
{"origin_lat": "31.489489", "origin_lng": "120.266144", "destination_lat": "31.499521", "destination_lng": "120.284699"}

**Tool Call/Response**

**[160]** *Tool Response:*

distance: 2.08 km, estimated travel time: 14 min

**Assistant**

[161] **Train Timing Rationale**

For your **outbound trip (Oct 24)**, three trains meet the ≤¥400 budget:

Train: D3142/D3143    Departure: 09:48    Arrival: 16:54    Price: ¥337    Notes: **Selected** – earliest arrival, maximizes your first evening in Wuxi

Train: D3136/D3137    Departure: 11:25    Arrival: 19:04    Price: ¥348    Notes: Arrives late evening, limiting dinner/activity options

Train: D2282    Departure: 14:42    Arrival: 22:10    Price: ¥348    Notes: Arrives very late, essentially wasting the first day

The **09:48 departure** was chosen because it provides a reasonable morning start from Fuzhou, arrives in Wuxi by late afternoon, and leaves enough time for hotel check-in, dinner, and a brief evening activity.

For the **return trip (Oct 27)**, four trains meet the budget:

Train: D2281    Departure: 09:09    Arrival: 16:35    Price: ¥300.5    Notes: **Selected** – earliest arrival back to Fuzhou, departs from convenient Wuxi Station

Train: D3135/D3138    Departure: 09:33    Arrival: 17:24    Price: ¥303.0    Notes: Slightly later arrival

Train: D3135/D3138    Departure: 09:43    Arrival: 17:24    Price: ¥299.0    Notes: Cheapest, but departs from **Wuxi New District Station** (less central)

Train: D3141/D3144    Departure: 12:23    Arrival: 19:28    Price: ¥348.0    Notes: Later departure & arrival

The **09:09 departure** offers the earliest return to Fuzhou while keeping you on a convenient morning schedule. The marginally cheaper option (¥299) uses a different station that would require extra travel time on your last morning.

**Conclusion:** The selected trains provide the best balance of **cost (≤¥400), schedule efficiency, and convenience**.

—

**Revised Itinerary with Closer Hotel**

I've switched your hotel to **Hanjin Hotel** (4.5 stars), located **2.75 km from the city center** (within your 3 km requirement). All transportation times have been recalculated based on the new hotel location.

—

**Day 1 (2025-10-24): Fuzhou → Wuxi**

Hotel: Hotel_00040914

   [09:33-09:48] Intercity Transportation (ID: Train_00001629)

      Description: Take train D3142/D3143 from Fuzhou South Railway Station to Wuxi Station. Arrive at the station 15 minutes before departure.

   [16:54-17:30] Local Transportation

      Description: Transfer from Wuxi Station to Hanjin Hotel (approx. 30 min).

   [17:30-17:45] Hotel Check-in

      Description: Check in at Hanjin Hotel (single room).

   [17:45-18:00] Local Transportation

      Description: Travel from hotel to Wang Steak barbecue restaurant (approx. 13 min).

   [18:00-19:00] Restaurant (ID: restaurant_3689704)

      Description: Dinner at Wang Steak (Barbecue, 4.5 stars). No solo set menu; order on site.

   [19:00-19:15] Local Transportation

      Description: Return from restaurant to hotel (approx. 13 min).

**Day 2 (2025-10-25): Wuxi**

. . . [omitted for clarity]

**Day 3 (2025-10-26): Wuxi**

. . . [omitted for clarity]

**Day 4 (2025-10-27): Wuxi → Fuzhou**

   [08:15-08:45] Local Transportation

      Description: Travel from hotel to Wuxi Station (approx. 30 min).

   [09:09-16:35] Intercity Transportation (ID: Train_00000845)

      Description: Take train D2281 from Wuxi Station to Fuzhou South Railway Station.

**User**

[162] Thanks for the detailed explanation on the trains, that makes sense. I have a few more points to add for the hotel and restaurants. First, I don't need breakfast included at the hotel, so please make sure the booking is for zero breakfasts. Second, I'd prefer restaurants where I can just walk in without needing to make a mandatory reservation in advance. Also, I'm bringing my dog along, so the hotel must be pet friendly. Lastly, could you clarify something? For the 'Ecology, Flora & Fauna Zones' priority, does a classical garden like Li Garden truly count as a primary 'Flora & Fauna Zone', or should the itinerary focus more on dedicated nature reserves or wildlife parks first?

**Assistant**

[163] *Tool Calls (1):*

Tool 1: search_attractions
**Arguments:**
{"city": "Wuxi", "attraction_name": "park", "rating": "4.0", "page_size": "10"}

**Tool Call/Response**

[164] *Tool Response:*

Showing 1-10 of 10 results.
POI ID: 61116204 | Wuxi | Wuxi Sunac Amusement Park | level: None | longitude: 120.271041, latitude: 31.450076 | distance: 5.98km
rating: 4.4 (4295 comments) | popularity Score: 5.8 | opening hours: 10:30 { 20:30 | reference visit time: 0.5-1 day | ticket price: 88.0
POI ID: 61116205 | Wuxi | Wuxi·Sunac Ocean Park | level: None | longitude: 120.272148, latitude: 31.448296 | distance: 6.06km
rating: 4.4 (2109 comments) | popularity Score: 5.2 | opening hours: 10:00 { 17:00 | reference visit time: 2-3 hours | ticket price: 180.0
POI ID: 10524099 | Wuxi | Wuxi Zoo,Taihu Lake Amusement Park | level: None | longitude: 120.235333, latitude: 31.584891 | distance: 12.69km
rating: 4.6 (3982 comments) | popularity Score: 4.7 | opening hours: 08:30 { 16:30 | reference visit time: 0.5-1 day | ticket price: 100.0
POI ID: 10536146 | Wuxi | Lihu Park | level: None | longitude: 120.269494, latitude: 31.520571 | distance: 5.19km
rating: 4.6 (134 comments) | popularity Score: 3.9 | opening hours: 00:00 { 24:00 | reference visit time: 3-4 hours | ticket price: Free
POI ID: 52285296 | Wuxi | Wuxi Xihui Park | level: None | longitude: 120.270922, latitude: 31.577575 | distance: 10.35km
rating: 4.5 (16 comments) | popularity Score: 3.6 | opening hours: 08:00 { 17:00 | reference visit time: 1-2 hours | ticket price: 70.0
POI ID: 87859 | Wuxi | E'bi Zui Park (Goose Nose Park) | level: None | longitude: 120.271458, latitude: 31.934248 | distance: 49.29km
rating: 4.4 (352 comments) | popularity Score: 3.1 | opening hours: 00:00 { 23:59 | reference visit time: 1-2 hours | ticket price: Free
POI ID: 91749 | Wuxi | Lihu Lake Central Park | level: None | longitude: 120.248784, latitude: 31.521683 | distance: 6.89km
rating: 4.5 (113 comments) | popularity Score: 3.0 | opening hours: 06:30 { 22:00 | reference visit time: 2-3 hours | ticket price: Free
POI ID: 10519872 | Wuxi | Xuelang Mountain Ecological Park | level: None | longitude: 120.260155, latitude: 31.464848 | distance: 5.71km
rating: 4.4 (481 comments) | popularity Score: 2.9 | opening hours: 07:30 { 19:00 | reference visit time: 0.5-1 day | ticket price: 35.0
POI ID: 39525057 | Wuxi | Jingui Park | level: None | longitude: 120.311146, latitude: 31.495871 | distance: 0.54km
rating: 4.8 (44 comments) | popularity Score: 2.6 | opening hours: 00:00 { 24:00 | reference visit time: 1-3 hours | ticket price: Free
POI ID: 10519612 | Wuxi | Zhongshan Park | level: None | longitude: 120.271892, latitude: 31.907716 | distance: 46.35km
rating: 4.6 (64 comments) | popularity Score: 2.2 | opening hours: 00:00 { 24:00 | reference visit time: 1-2 hours | ticket price: Free

**Assistant**

**[165]** *Tool Calls (1):*
Tool 1: get_attraction_detail_with_products
**Arguments:**
{"poi_id": "10519872"}

**Tool Call/Response**

**[166]** *Tool Response:*
POI ID: 10519872 | Wuxi | Xuelang Mountain Ecological Park | level: N/A | longitude: 120.260155, latitude: 31.464848
categories: Ecology, Flora & Fauna Zones, Family & Kids Friendly, Outdoor Adventure & Sports
rating: 4.4 (481 comments) | popularity score: 2.9 | opening hours: 07:30 { 19:00 | reference visit time: 0.5-1 day | features: Agricultural production, popular science education
product_id: 10519872_1 | Adult | price: 35.0
product_id: 10519872_2 | Student | price: 24.5
product_id: 10519872_3 | Senior | price: 21.0
product_id: 10519872_4 | Child | price: 17.5
product_id: 10519872_5 | Family Package (2 Adults + 1 Child) | price: 77.0

**Assistant**

**[167]** *Tool Calls (1):*
Tool 1: get_attraction_detail_with_products
**Arguments:**
{"poi_id": "10536146"}

**Tool Call/Response**

**[168]** *Tool Response:*
POI ID: 10536146 | Wuxi | Lihu Park | level: N/A | longitude: 120.269494, latitude: 31.520571
categories: Natural Scenery, City Sightseeing & Landmarks, Photo Spots & Instagrammable, Leisure, Wellness & Resort, Family & Kids Friendly
rating: 4.6 (134 comments) | popularity score: 3.9 | opening hours: 00:00 { 24:00 | reference visit time: 3-4 hours | features: None
product_id: 10536146_1 | Free Ticket | price: 0

**Omitted Turns**

**Turns [169–174] omitted for brevity**

Intermediate tool calls/responses suppressed

**Assistant**

**[175]** *Tool Calls (1):*
Tool 1: get_route_estimate
**Arguments:**
{"origin_lat": "31.514576", "origin_lng": "120.279077", "destination_lat": "31.489207", "destination_lng": "120.299003"}

**Tool Call/Response**

**[176]** *Tool Response:*
distance: 3.39 km, estimated travel time: 17 min

---

**Assistant**

**[177]** *Tool Calls (1):*

Tool 1: get_route_estimate

**Arguments:**

{"origin_lat": "31.489207", "origin_lng": "120.299003", "destination_lat": "31.499521", "destination_lng": "120.284699"}

---

**Tool Call/Response**

**[178]** *Tool Response:*

distance: 1.78 km, estimated travel time: 14 min

---

**Assistant**

**[179]** **Clarification on Ecology, Flora & Fauna Zones**

You raise a valid point. A classical garden like Li Garden, while containing flora, is not a dedicated **ecological zone or wildlife park**. To better match your priority, I have replaced Li Garden with **Xuelang Mountain Ecological Park** – an attraction that explicitly lists "Ecology, Flora & Fauna Zones" as its primary category. This is a dedicated ecological park with agricultural production and education, offering a more authentic nature experience.

**Other Adjustments Made:**

1. **Pet-friendly hotel** – Hanjin Hotel ('Hotel_00040914') is marked 'pet_friendly: True'.

2. **Zero-breakfast room** – The selected product 'P_H_00760382' has 'breakfast_num: 0'.

3. **No mandatory reservations** – The Korean restaurant on Day 2 ('restaurant_7471268') required a mandatory reservation. It has been replaced with **Zhang Liang malatang** ('restaurant_3903050'), which has 'must_reserve: False' (5.0 stars, Korean cuisine).

**Updated Itinerary**

**Day 1 (2025-10-24): Fuzhou → Wuxi**

Hotel: Hotel_00040914

  [09:33-09:48] Intercity Transportation (ID: Train_00001629)

    Description: Take train D3142/D3143 from Fuzhou South Railway Station to Wuxi Station. Arrive at the station 15 minutes before departure.

  [16:54-17:30] Local Transportation

    Description: Transfer from Wuxi Station to Hanjin Hotel (approx. 30 min).

  [17:30-17:45] Hotel Check-in

    Description: Check in at Hanjin Hotel (single room, no breakfast).

  [17:45-18:00] Local Transportation

    Description: Travel from hotel to Wang Steak barbecue restaurant (approx. 13 min).

  [18:00-19:00] Restaurant (ID: restaurant_3689704)

    Description: Dinner at Wang Steak (Barbecue, 4.5 stars). No solo set menu; order on site. No mandatory reservation required.

  [19:00-19:15] Local Transportation

    Description: Return from restaurant to hotel (approx. 13 min).

**Day 2 (2025-10-25): Wuxi**

...[omitted for clarity]

**Day 3 (2025-10-26): Wuxi**

...[omitted for clarity]

**Day 4 (2025-10-27): Wuxi → Fuzhou**

  [08:15-08:45] Local Transportation

    Description: Travel from hotel to Wuxi Station (approx. 30 min).

  [09:09-16:35] Intercity Transportation (ID: Train_00000845)

    Description: Take train D2281 from Wuxi Station to Fuzhou South Railway Station.

**User**

**[180]** Thanks for making those changes, the new ecological park and the pet-friendly, no-reservation options look much better. I've been thinking about a few more specifics. For the return flight, please make sure you don't book with 9 Air. Also, when picking restaurants, I'd really prefer ones that have a lot of reviews, like at least 1000, as they seem more reliable. And for the hotel, can you ensure it has at least 100 reviews as well? Oh, and I changed my mind about breakfast - I'd actually like the hotel to include at least one breakfast for me.

**Assistant**

**[181]** *Tool Calls (1):*

Tool 1: search_hotels

**Arguments:**

{"city": "Wuxi", "check_in_date": "2025-10-24", "check_out_date": "2025-10-27", "longitude": "120.311889", "latitude": "31.491064", "distance_threshold": "3", "is_pet_friendly": "true", "review_count": "100", "sort_key": "stars", "sort_order": "desc", "page_size": "10"}

**Tool Call/Response**

**[182]** *Tool Response:*

Showing 1-4 of 4 results.
Hotel_id: Hotel_00041418 | Worldhotel Grand Juna Wuxi | type: Luxury | price: 510.6+ | stars: 5.0/5 |
review_count: 721 | longitude: 120.320327, latitude: 31.477452 | distance: 1.71km
Hotel_id: Hotel_00040914 | Hanjin Hotel | type: Midscale | price: 367.47+ | stars: 4.5/5 | review_count: 200 |
longitude: 120.284699, latitude: 31.499521 | distance: 2.75km
Hotel_id: Hotel_00040977 | Wuxi Kailai Hotel | type: Luxury | price: 820.1+ | stars: 4.5/5 | review_count: 254 |
longitude: 120.327029, latitude: 31.504266 | distance: 2.05km
Hotel_id: Hotel_00041318 | Wuxi Zhanmei Boutique Hotel | type: Upscale | price: 402.85+ | stars: 4.0/5 |
review_count: 274 | longitude: 120.295741, latitude: 31.487359 | distance: 1.59km

**Assistant**

**[183]** *Tool Calls (1):*

Tool 1: get_hotel_detail_with_products

**Arguments:**

{"hotel_id": "Hotel_00041418", "check_in_date": "2025-10-24", "check_out_date": "2025-10-27", "room_num": "1", "person_num": "1", "page_size": "10"}

**Tool Call/Response**

**[184]** *Tool Response:*

Hotel_id: Hotel_00041418 | Wuxi | type: Luxury | stars: 5.0/5 | review_count: 721.0 | longitude: 120.320327 |
latitude: 31.477452 | product_rating: 9.0 | environment_rating: 9.1 | service_rating: 9.2 |pet_friendly: True
Showing 1-10 of 12 products.
product_id: P_H_00769371 | room_type: single | maximum occupancy per room: 1 | breakfast_num: 0 | cancel_policy:
3 (Non-cancellable) | has_window: False | avg_price_per_night_per_room: 510.6 |
product_id: P_H_00769372 | room_type: single | maximum occupancy per room: 1 | breakfast_num: 1 | cancel_policy:
3 (Non-cancellable) | has_window: True | avg_price_per_night_per_room: 582.19 |
product_id: P_H_00769373 | room_type: large | maximum occupancy per room: 2 | breakfast_num: 0 | cancel_policy:
3 (Non-cancellable) | has_window: False | avg_price_per_night_per_room: 612.72 |
product_id: P_H_00769374 | room_type: large | maximum occupancy per room: 2 | breakfast_num: 1 | cancel_policy:
3 (Non-cancellable) | has_window: False | avg_price_per_night_per_room: 626.09 |
product_id: P_H_00769375 | room_type: large | maximum occupancy per room: 2 | breakfast_num: 2 | cancel_policy:
3 (Non-cancellable) | has_window: False | avg_price_per_night_per_room: 639.45 |
product_id: P_H_00769378 | room_type: double | maximum occupancy per room: 2 | breakfast_num: 2 | cancel_policy:
3 (Non-cancellable) | has_window: False | avg_price_per_night_per_room: 792.63 |
product_id: P_H_00769376 | room_type: double | maximum occupancy per room: 2 | breakfast_num: 0 | cancel_policy:
3 (Non-cancellable) | has_window: True | avg_price_per_night_per_room: 851.0 |
product_id: P_H_00769377 | room_type: double | maximum occupancy per room: 2 | breakfast_num: 1 | cancel_policy:
3 (Non-cancellable) | has_window: True | avg_price_per_night_per_room: 865.85 |
product_id: P_H_00769379 | room_type: family | maximum occupancy per room: 3 | breakfast_num: 0 | cancel_policy:

3 (Non-cancellable) | has_window: True | avg_price_per_night_per_room: 1021.2 |
product_id: P_H_00769380 | room_type: family | maximum occupancy per room: 3 | breakfast_num: 1 | cancel_policy:
3 (Non-cancellable) | has_window: True | avg_price_per_night_per_room: 1036.05 |

---

**Assistant**

**[185]** *Tool Calls (1):*
Tool 1: search_restaurants
**Arguments:**
{"city": "Wuxi", "categories": "[\"Korean Cuisine\"]", "stars": "4.5", "review_count": "1000", "sort_key":
"review_count", "sort_order": "desc", "page_size": "10"}

---

**Tool Call/Response**

**[186]** *Tool Response:*
Showing 1-3 of 3 results.
Restaurant_id: restaurant_6455324 | Wang Steak | category: Korean Cuisine | avg_price: 141.0 | stars: 4.5/5 |
review_count: 2236 | open_hours: 11:00-22:00 | longitude: 120.281903, latitude: 31.512466 | distance: 3.71km
Restaurant_id: restaurant_7471268 | Korean Bucket House BBQ Restaurant | category: Korean Cuisine | avg_price:
160.5 | stars: 5.0/5 | review_count: 2013 | open_hours: 11:00-22:30 | longitude: 120.296028, latitude:
31.487779 | distance: 1.55km
Restaurant_id: restaurant_7832802 | Find Charcoal Barbecue · Original Seoul Charcoal barbecue | category:
Korean Cuisine | avg_price: 151.5 | stars: 4.5/5 | review_count: 1140 | open_hours: 11:00-23:00 | longitude:
120.284334, latitude: 31.5157 | distance: 3.78km

---

**Omitted Turns**

**Turns [187–204] omitted for brevity**
Intermediate tool calls/responses suppressed

---

**Assistant**

**[205]** *Tool Calls (1):*
Tool 1: get_route_estimate
**Arguments:**
{"origin_lat": "31.514576", "origin_lng": "120.279077", "destination_lat": "31.512466", "destination_lng":
"120.281903"}

---

**Tool Call/Response**

**[206]** *Tool Response:*
distance: 0.36 km, estimated travel time: 4 min

---

**Assistant**

**[207]** *Tool Calls (1):*
Tool 1: get_route_estimate
**Arguments:**
{"origin_lat": "31.525381", "origin_lng": "120.272061", "destination_lat": "31.48857", "destination_lng":
"120.30286"}

---

**Tool Call/Response**

**[208]** *Tool Response:*
distance: 5.02 km, estimated travel time: 20 min

**Assistant**

**[209]**

**Clarification on Transportation**

Your itinerary uses **trains**, not flights. There are no flights between Fuzhou and Wuxi in the system for your dates, so no airline (including 9 Air) is involved. The selected trains are the most cost- and time-effective options meeting your <¥400 one-way budget.

**Changes Made to Meet Your New Requests**

Request: **Hotel with ≥100 reviews & breakfast**    How Addressed: Changed to **Worldhotel Grand Juna Wuxi** (5 stars, 721 reviews, pet-friendly). Room product includes **1 breakfast** per night.

Request: **Restaurants with ≥1000 reviews, no mandatory reservation**    How Addressed: Day 1: Wang Steak (Barbecue, 1200 reviews) – already met. Day 2: Wang Steak (Korean Cuisine, 2236 reviews) – solo set menu available, no mandatory reservation. Day 3: Oven fish (Barbecue, 4579 reviews) – no mandatory reservation.

Request: **Home-style Cooking requirement**    How Addressed: Dropped because no restaurant in this category has ≥1000 reviews. Replaced with high-review Barbecue.

Request: **Pet-friendly hotel**    How Addressed: Worldhotel Grand Juna Wuxi is marked 'pet_friendly: True'.

**Updated Itinerary**

All transportation times have been recalculated from the new hotel location.

**Day 1 (2025-10-24): Fuzhou → Wuxi**

Hotel: Hotel_00041418

 [09:33-09:48] Intercity Transportation (ID: Train_00001629)

  Description: Take train D3142/D3143 from Fuzhou South Railway Station to Wuxi Station. Arrive at the station 15 minutes before departure.

 [16:54-17:30] Local Transportation

  Description: Transfer from Wuxi Station to Worldhotel Grand Juna Wuxi (approx. 32 min).

 [17:30-17:45] Hotel Check-in

  Description: Check in at Worldhotel Grand Juna Wuxi (single room with breakfast).

 [17:45-18:06] Local Transportation

  Description: Travel from hotel to Wang Steak barbecue restaurant (approx. 21 min).

 [18:06-19:06] Restaurant (ID: restaurant_3689704)

  Description: Dinner at Wang Steak (Barbecue, 4.5 stars, 1200 reviews). No solo set menu; order on site. No mandatory reservation.

 [19:06-19:27] Local Transportation

  Description: Return from restaurant to hotel (approx. 21 min).

**Day 2 (2025-10-25): Wuxi**

. . . [omitted for clarity]

**Day 3 (2025-10-26): Wuxi**

. . . [omitted for clarity]

**Day 4 (2025-10-27): Wuxi → Fuzhou**

 [08:15-08:47] Local Transportation

  Description: Travel from hotel to Wuxi Station (approx. 32 min).

 [09:09-16:35] Intercity Transportation (ID: Train_00000845)

  Description: Take train D2281 from Wuxi Station to Fuzhou South Railway Station.

**User**

**[210]** Thanks for the updates, the new hotel and restaurants look great. I have one more request for the return trip: I'd like the train to depart after 08:00. Also, could you clarify something for me? The itinerary mentions a 'solo set menu' at the Korean restaurant on Day 2. What does that actually include? Is it a fixed course meal, and does it offer good value compared to ordering à la carte?

**Assistant**

**[211]**

**Clarifications**

1. **Return train timing**: Your return train (D2281) departs at **09:09**, which is after 08:00, so this requirement is already satisfied.

2. **Solo set menu at Wang Steak (Korean Cuisine)**:

- **What it includes**: The product 'Pfd09d3861615' is a **fixed-course meal for one person** priced at **¥132.68**. While the exact dishes are not listed in the database, set menus at Korean barbecue restaurants typically include a selection of marinated meats, side dishes (banchan), rice, soup, and sometimes a beverage.

- **Value comparison**: The average price per person at this restaurant is ¥141.00. The set menu costs ¥132.68, which is **about 6% below the average**, suggesting it offers a slight discount compared to ordering the same items à la carte. Moreover, a set menu ensures a balanced meal without the hassle of choosing individual dishes. Since your request for a return departure after 08:00 is already met, no modifications to the itinerary are needed.

**User**

**[212]** Alright, thanks for the clarification on the set menu. I noticed the itinerary says the daily schedule is empty, but I can see the detailed plan you provided in your response. Is there a technical issue on my end, or was that just a note at the end of your message? Also, regarding the return train timing, I'd like to update my preference: please make sure the departure is after 10:00, not 08:00. Finally, you mentioned earlier that some train timings might be inconsistent. Could you double-check that the arrival and departure times for both the inbound and outbound trains in this plan are accurate and properly aligned?

**Assistant**

**[213]** *Tool Calls (1):*

Tool 1: search_trains

**Arguments:**

{"departure_city": "Wuxi", "arrival_city": "Fuzhou", "date_str": "2025-10-27", "sort_key": "time", "sort_order": "asc", "page_size": "10"}

**Tool Call/Response**

**[214]** *Tool Response:*

Showing 1-4 of 4 results.
Train_id: Train_00000845 | D2281 | 9:09-16:35 | Wuxi Station → Fuzhou South Railway Station | minimum price: 300.5
Train_id: Train_00001517 | D3135/D3138 | 9:33-17:24 | Wuxi Station → Fuzhou South Railway Station | minimum price: 303.0
Train_id: Train_00001523 | D3135/D3138 | 9:43-17:24 | Wuxi New District Station → Fuzhou South Railway Station | minimum price: 299.0
Train_id: Train_00001594 | D3141/D3144 | 12:23-19:28 | Wuxi Station → Fuzhou South Railway Station | minimum price: 348.0

**Assistant**

**[215]** *Tool Calls (1):*

Tool 1: get_train_detail_with_products

**Arguments:**

{"train_id": "Train_00001594", "date_str": "2025-10-27"}

**Tool Call/Response**

**[216]** *Tool Response:*

Train_id: Train_00001594 | D3141/D3144 | 12:23-19:28 | Wuxi Station → Fuzhou South Railway Station
product_id: P_T_00018969 | Second class | ctrip | price: 348.0 |
product_id: P_T_00018970 | Second class | alitrip | price: 348.0 |

```
product_id: P_T_00018971 | Second class | qunar | price: 348.0 |
product_id: P_T_00018972 | Second class | direct | price: 348.0 |
```

**Assistant**

**[217]** *Tool Calls (1):*
Tool 1: get_train_detail_with_products
**Arguments:**
{"train_id": "Train_00001629", "date_str": "2025-10-24"}

**Tool Call/Response**

**[218]** *Tool Response:*
```
Train_id: Train_00001629 | D3142/D3143 | 9:48-16:54 | Fuzhou South Railway Station → Wuxi Station
product_id: P_T_00019389 | Second class | ctrip | price: 337.0 |
product_id: P_T_00019390 | Second class | alitrip | price: 337.0 |
product_id: P_T_00019391 | Second class | qunar | price: 337.0 |
product_id: P_T_00019392 | Second class | direct | price: 337.0 |
```

**Assistant**

**[219]** *Tool Calls (1):*
Tool 1: get_attraction_detail_with_products
**Arguments:**
{"poi_id": "39525057"}

**Tool Call/Response**

**[220]** *Tool Response:*
```
POI ID: 39525057 | Wuxi | Jingui Park | level: N/A | longitude: 120.311146, latitude: 31.495871
categories: Natural Scenery, Leisure, Wellness & Resort, Photo Spots & Instagrammable, Family & Kids Friendly
rating: 4.8 (44 comments) | popularity score: 2.6 | opening hours: 00:00 { 24:00 | reference visit time: 1-3
hours | features: Construction highlights the characteristics of blending mountains and rivers
product_id: 39525057_1 | Free Ticket | price: 0
```

**Assistant**

**[221]** *Tool Calls (1):*
Tool 1: get_route_estimate
**Arguments:**
{"origin_lat": "31.477452", "origin_lng": "120.320327", "destination_lat": "31.495871", "destination_lng":
"120.311146"}

**Tool Call/Response**

**[222]** *Tool Response:*
distance: 2.22 km, estimated travel time: 14 min

**Assistant**

**[223]** *Tool Calls (1):*
Tool 1: get_route_estimate
**Arguments:**
{"origin_lat": "31.495871", "origin_lng": "120.311146", "destination_lat": "31.588048", "destination_lng":
"120.306376"}

**[224]** *Tool Response:*

`distance: 10.23 km, estimated travel time: 30 min`

**Assistant**

**[225] Day 1 (2025-10-24): Fuzhou → Wuxi**

Hotel: Hotel_00041418

 [09:33-09:48] Intercity Transportation (ID: Train_00001629)

  Description: Take train D3142/D3143 from Fuzhou South Railway Station to Wuxi Station. Arrive at the station 15 minutes before departure.

 [16:54-17:30] Local Transportation

  Description: Transfer from Wuxi Station to Worldhotel Grand Juna Wuxi (approx. 32 min).

 [17:30-17:45] Hotel Check-in

  Description: Check in at Worldhotel Grand Juna Wuxi (single room with breakfast).

 [17:45-18:06] Local Transportation

  Description: Travel from hotel to Wang Steak barbecue restaurant (approx. 21 min).

 [18:06-19:06] Restaurant (ID: restaurant_3689704)

  Description: Dinner at Wang Steak (Barbecue, 4.5 stars, 1200 reviews). No solo set menu; order on site. No mandatory reservation.

 [19:06-19:27] Local Transportation

  Description: Return from restaurant to hotel (approx. 21 min).

**Day 2 (2025-10-25): Wuxi**

Hotel: Hotel_00041418

 [08:00-08:22] Local Transportation

  Description: Travel from hotel to Xuelang Mountain Ecological Park (approx. 22 min).

 [08:30-12:00] Attraction (ID: 10519872)

  Description: Visit Xuelang Mountain Ecological Park (dedicated Ecology, Flora & Fauna Zones, 3.5 hours).

 [12:00-12:26] Local Transportation

  Description: Travel from Xuelang Mountain to Best Scenic Spot of Taihu Lake (approx. 26 min).

 [12:30-15:00] Attraction (ID: 56807028)

  Description: Visit Best Scenic Spot of Taihu Lake (natural scenery, 2.5 hours).

 [15:00-15:28] Local Transportation

  Description: Travel from Taihu Lake to Wuxi Museum (approx. 28 min).

 [15:30-17:00] Attraction (ID: 86783)

  Description: Visit Wuxi Museum (cultural attraction, 1.5 hours).

 [17:00-17:19] Local Transportation

  Description: Travel from museum to Wuxi Grand Theatre (approx. 19 min).

 [17:20-19:20] Attraction (ID: 18567012)

  Description: Visit Wuxi Grand Theatre (nightlife & live shows, 2 hours).

 [19:20-19:24] Local Transportation

  Description: Travel from theatre to Wang Steak Korean restaurant (approx. 4 min).

 [19:30-20:30] Restaurant (ID: restaurant_6455324)

  Description: Dinner at Wang Steak (Korean Cuisine, 4.5 stars, 2236 reviews). Solo set menu included. No mandatory reservation.

 [20:30-20:51] Local Transportation

  Description: Return from restaurant to hotel (approx. 21 min).

**Day 3 (2025-10-26): Wuxi**

Hotel: Hotel_00041418

 [08:30-09:04] Local Transportation

  Description: Travel from hotel to Wuxi Zoo, Taihu Lake Amusement Park (approx. 34 min).

 [09:10-16:40] Attraction (ID: 10524099)

  Description: Full day at Wuxi Zoo (wildlife park, ecology & fauna). Lunch will be handled inside the attrac-

tion.

    [16:40-17:05] Local Transportation

      Description: Travel from zoo to Star of Taihu Ferris Wheel (approx. 25 min).

    [17:10-18:10] Attraction (ID: 33190543)

      Description: Visit Star of Taihu Ferris Wheel (night views & night tours, 0.5-1 hour).

    [18:10-18:30] Local Transportation

      Description: Travel from ferris wheel to Oven fish restaurant (approx. 20 min).

    [18:30-19:30] Restaurant (ID: restaurant_9134254)

      Description: Dinner at Oven fish (Barbecue, 4.5 stars, 4579 reviews). No solo set menu; order on site. No mandatory reservation.

    [19:30-19:44] Local Transportation

      Description: Return from restaurant to hotel (approx. 14 min).

**Day 4 (2025-10-27): Wuxi → Fuzhou**

    [08:30-09:00] Hotel Check-in

      Description: Breakfast at hotel and check out.

    [09:00-09:14] Local Transportation

      Description: Travel from hotel to Jingui Park (approx. 14 min).

    [09:15-10:45] Attraction (ID: 39525057)

      Description: Visit Jingui Park (natural scenery, free, 1.5 hours).

    [10:45-11:15] Local Transportation

      Description: Travel from Jingui Park to Wuxi Station (approx. 30 min).

    [12:23-19:28] Intercity Transportation (ID: Train_00001594)

      Description: Take train D3141/D3144 from Wuxi Station to Fuzhou South Railway Station.

