# OpenReview forum: "TRIP-Bench: A Benchmark for Long-Horizon Interactive Agents in Real-World Scenarios"
_ICML.cc/2026/Conference — ICML 2026 regular_

### Official Review · Reviewer_UTHf · 2026-03-06

**Soundness:** 2
**Presentation:** 3
**Significance:** 3
**Originality:** 3
**Overall Recommendation:** 4
**Confidence:** 4

**Summary:**

TRIP-Bench is a long-horizon interactive benchmark for real-world travel planning scenarios, designed to evaluate constraint satisfaction, tool-use coordination, plan consistency, and adaptation to evolving preferences in multi-turn dialogues. The paper also proposes GTPO, an online RL method, and benchmarks it against prior approaches. Experimental results show that mainstream models struggle under stricter and harder settings, while GTPO yields consistent gains, highlighting both the benchmark's difficulty and its utility for driving new methods.

**Compliance With Llm Reviewing Policy:**

Affirmed.

**Final Justification:**

My concerns have been largely addressed, especially regarding the failure attribution for zero-score cases. I maintain the current positive score.

**Key Questions For Authors:**

1. What are the primary causes of the zero-score cases in Table 2?
2. Could the authors provide any proxy evaluation of GTPO on hard subsets?
3. Could the authors report GTPO training/inference cost and latency metrics?

**Limitations:**

Yes

**Strengths And Weaknesses:**

**Strengths:**
1. The benchmark pipeline is well-designed and possesses a relatively comprehensive logical flow for data generation/validation.
2. The GTPO method is clearly written, featuring formulas, component motivations, and ablation studies.
3. The paper is easy to follow, and the figures are clear.

**Weaknesses:**
1. The failure attribution for zero-score cases is under-analyzed. The paper attributes low aggregate performance to benchmark difficulty and strict metrics, but does not provide a fine-grained study for the zero/near-zero entries in Table 2. Without separating genuine planning failures from harness/evaluation issues (e.g., tool-call format mismatches, timeouts/context truncation, or brittle parsing), the results are harder to interpret, and failure attribution is less reliable.

2. Despite the main methodological contribution, GTPO is not evaluated on the hard subset due to context-length constraints, with results limited to easy/mid splits. This weakens the claim that GTPO addresses the benchmark's core challenge (long-horizon, multi-turn planning) and leaves open whether the reported gains persist under the most demanding settings.

3. The computational efficiency is not sufficiently reported. While Appendix D.1/Table 6 lists hyperparameters and system setup, the paper does not report concrete efficiency metrics for GTPO (e.g., training time, throughput, or inference latency). This limits the assessment of practicality and deployment for a method aimed at long-horizon interactive agents.

---

> ### Author Rebuttal · Authors · 2026-03-29
>
> ### Weakness 1 & Question 1
>
> Zero or near-zero outcomes are not caused by any single factor. To better distinguish genuine planning failures from issues in the runtime or evaluation pipeline, we further examined performance on the hard subsets for both DS and GPT, and categorized errors into three types: general-only failure, user-only failure, and mixed failure. The results show that about **82%** of the samples involve failures on both general constraints and user constraints, while only **18%** are attributable to a single type of failure. This suggests that, in the hard setting, the typical problem is not simply missing one user preference, but rather a simultaneous breakdown in feasibility, planning quality, and satisfaction of user requirements as interaction complexity increases.
>
> More specifically, among these samples, we observed an average of 2.9 unmet general constraints and 2.8 unmet user requirements per sample. After grouping the failed checks into the natural-language subcategories defined in Appendix B.4, we find that the dominant source of failures on the general side is **Temporal reasonableness**: errors related to opening hours, daily scheduling, activity duration, local transportation, and intercity transportation connections account for **56%** of all failed general checks. Another 29% of failed general checks fall under Basic feasibility, including structural validity, POI validity, and information completeness issues.
>
> We also analyzed the distribution of unmet user requirements. The most common failures arise from more complex requirements, such as **global constraints** and preference prioritization, as illustrated in Figure 4.
>
> Finally, due to the **200k context window limitation** of the GLM model, around **20%** of its cases in the hard subset exceed the context length limit, whereas this issue does not occur for the other models. Across all trajectories, we found only two cases of tool-calling failure, indicating that for current strong models, correct tool usage is no longer a major bottleneck.
>
> ### Weakness 2 & Question 2
>
> We evaluate under a setting where the dialogue is truncated to **128k tokens**, and each example is cut at the last turn before exceeding the context limit.
>
> Under this setting, the 14B model achieves an average loose / strict pass rate of 8.5% / 0% on the hard subset, while the 32B model reaches **16% / 2%**, respectively. For reference, under the same 128k setting, Gemini 3 Pro achieves 14% / 1%. These results show that even under the most challenging hard setting, **GTPO still provides consistent gains**, and the 32B model already surpasses this strong baseline on the strict metric.
>
> ### Weakness 3 & Question 3
>
> All experiments were trained on **32 H200 GPUs**. For the 14B model, each RL training step takes approximately 20 minutes. During inference, we use 8 H200 GPUs; under the 128k context length setting, the average inference time is about **5 minutes per task**. For the 32B model, both training and inference costs are approximately **1.5×** those of the 14B model.

---

> > ### Author Rebuttal · Reviewer_UTHf · 2026-04-02
> >
> > Thank you for the responses. My concerns have been largely addressed, especially regarding the failure attribution for zero-score cases. I maintain the current positive score.

---

### Official Review · Reviewer_cQHj · 2026-03-12

**Soundness:** 2
**Presentation:** 3
**Significance:** 3
**Originality:** 3
**Overall Recommendation:** 4
**Confidence:** 3

**Summary:**

This paper introduces TRIP-Bench, a benchmark designed to evaluate the performance of agents in long-horizon, interactive travel planning scenarios. The core value of this benchmark lies in its simulation of real-world complexity, particularly through the introduction of global constraints, modification-chains, and diverse user attributes and behaviors. These features enable a deep assessment of an agent's ability to satisfy complex constraints over extended multi-turn interactions.
TRIP-Bench provides 18 tools and over 40 types of travel requirements, with datasets categorized into three levels of difficulty. Notably, the hard split includes four additional evaluation sets specifically designed for targeted challenges. Furthermore, TRIP-Bench supports dialogues spanning up to 15 user turns, involving more than 150 tool calls and context lengths that can exceed 200,000 tokens. Experimental results demonstrate that even the current state-of-the-art models perform poorly when faced with the challenges presented by this benchmark.
Additionally, the authors propose GTPO (Group Relative Turn-level Policy Optimization), an online multi-turn reinforcement learning method featuring global Instruction reward normalization, turn-level reward normalization and turn-wise reward differencing. Experimental results indicate that the proposed GTPO method improves constraint satisfaction and interaction robustness compared to SFT and standard GRPO baselines.

**Compliance With Llm Reviewing Policy:**

Affirmed.

**Final Justification:**

Thank the authors for the rebuttal. I will keep my score.

**Key Questions For Authors:**

1. Reproducibility of the Evaluation Pipeline
The reliability of TRIP-Bench highly depends on the evaluation pipeline. The authors are requested to provide more detailed specifications regarding dataset construction, such as the exact prompting strategy and model configuration used for the generator and validator.
2. Relationship Between Long-Horizon Reasoning and Constraint Satisfaction
Although the benchmark is stated to evaluate long-horizon Interaction capabilities, the main construction and evaluation are related to satisfying multiple constraints simultaneously and diverse user attributes and behaviors. Do the authors plan to design dedicated metrics or ablation studies to explicitly measure model performance in planning depth, reasoning horizon, and multi-round tool invocation?
3. Potential Bias in the User Simulator
The user simulator is built upon DeepSeek models. Have the authors analyzed whether the intelligent agents based on DeepSeek might be significantly superior to other models due to their high compatibility with the user simulator? If so, what measures have been taken to mitigate this potential bias, or do the authors plan to release ablation studies on the backbone model of the user simulator?
4. Inconsistency in Figure 4
The results for “hotel cost” and “General: Local Transport” in Figure 4 appear to contradict the paper’s conclusions regarding local vs. global constraints. Could the authors explain the cause of this apparent contradiction?

**Limitations:**

Yes, the authors adequately discussed the limitations and potential negative societal impact of their work.

**Strengths And Weaknesses:**

Strengths:
1. The benchmark construction methodology is rigorous and reliable, encompassing hierarchical rubric-to-constraint generation, modification chain construction, and multi-level validation. The proposed GTPO method provides clear mathematical formulations that theoretically address specific challenges in multi-turn interactions, such as unifying rewards across different turns and diverse constraints.
2. The paper exhibits a clear overall structure with well-defined sections. It thoroughly discusses the novelty compared to prior work and provides detailed explanations of the benchmark construction and evaluation, as well as the principles and algorithms of the proposed GTPO method.
3. This work integrates complex constraint satisfaction with dynamic multi-turn user interactions. It specifically evaluates the capability of agents to satisfy complex constraints in long-horizon, multi-turn interactions—a capability essential for real-world agent deployment and sustained human-agent interaction, yet notably underrepresented in existing benchmarks.
4. The experimental section encompasses evaluations across multiple models and various difficulty levels, revealing important insights regarding the limitations of current models and providing valuable implications for handling complex constraints.

Weaknesses:
1. Clarification of “Real-World” Claims
The paper emphasizes real-world scenarios and real-world data, while also stating that a large-scale, tool-augmented simulated environment is constructed. Explicit definitions of each term, along with their respective roles and significance in the proposed benchmark, would greatly improve reader understanding.
2. Insufficient Details on Dataset Construction
Section 3.2 introduces TRIP-Bench construction but does not clearly disclose the sources of candidate itineraries. More importantly, critical details of the evaluation pipeline—including the exact configurations of the generator and validator, prompting strategies and the validation procedures—are not fully described. Since the reliability of the benchmark strongly depends on the evaluation pipeline, which also affects the reward computation of the GTPO algorithm, these omissions compromise the reproducibility of the results.
3. Misalignment Between Claimed and Measured Capabilities
Although the benchmark is positioned as an evaluation for long-horizon agents, the evaluation focuses primarily on constraint satisfaction under multiple requirements rather than explicitly measuring reasoning depth, planning horizon, or tool collaboration. It remains unclear whether the benchmark truly assesses the multiple capabilities involved in long-horizon reasoning or merely tests complex constraint satisfaction. The relationship among complex constraint satisfaction, adaptation to diverse user behaviors, and long-horizon reasoning and planning needs explicit justification. Alternatively, the authors should either redesign the metrics to directly analyze planning behavior or redefine the purpose of the benchmark.
4. Limited Discriminative Power of Metrics
The success rates of many models are close to zero, especially under the strict evaluation setting. While high difficulty can be valuable, extremely low success rates limit the benchmark’s ability to meaningfully distinguish between different levels of model capability.
5. Insufficient Baselines for GTPO
The proposed GTPO method is compared against only a limited set of baselines, mainly SFT and GRPO. Including stronger and more diverse reinforcement learning baselines would better support the empirical claims regarding the superiority of the proposed method.
6. Potential Evaluation Bias
The user simulator is implemented based on DeepSeek models. This setup may introduce potential bias: DeepSeek-family models could interact more effectively with the simulator due to alignment or architectural similarities, which may undermine the fairness of cross-model comparisons.
7. Presentation Ambiguities and Errors
- Text–figure inconsistency: Line 158 states that “…… curate over 80 diverse natural language expressions”. When combined with Figure 1, this phrasing creates ambiguity as to whether this refers to more than 80 expressions per category or more than 80 total across 40 categories.
- Figure legend error: Figure 6 contains a clear legend error, in which all data series are incorrectly labeled as related to GRPO.

---

> ### Author Rebuttal · Authors · 2026-03-29
>
> ### Weakness 1
>
> Travel planning is a realistic task with practical constraints, and our benchmark is grounded in real-world data. **Tool-enhanced** means the model can actively retrieve, filter, and combine information through a relatively complete toolset (Appendix B.1), rather than relying on passively provided context. This enables active information acquisition for planning. **Simulated environment** means the evaluation is conducted offline for reproducibility and controllability, while still using real-world data and tool mechanisms.
>
> ### Weakness 2 & Question 1
>
> Candidate itineraries are mainly synthesized by combining cities, as described in Meta-information Synthesis. For each itinerary, constraints are added through a generate–verify–compose pipeline, followed by quality control to filter invalid or inconsistent itinerary–requirement pairs.
>
> Both the generator and validator are implemented as **deterministic code**, so no randomness-induced variance is introduced. The generator produces feasible candidate options, corresponding ID lists, and textual variants of the same requirement for later composition, while the validator checks whether a selected POI satisfies the requirement. The overall workflow is illustrated in the left part of Figure 1.
>
> The prompts for the agent and user simulator are already included in Appendix F. In the revision, we will add more fine-grained implementation details to the appendix. The dataset and code will be released upon acceptance.
>
> ### Weakness 3 & Question 2
>
> TRIP-Bench is not designed to evaluate only long-horizon reasoning. It measures a combination of long-horizon tasks, complex rules, multi-turn interaction, and reasoning-based planning.
>
> We agree that the final metric is mainly constraint satisfaction. We use it because it is objective, scalable, and consistently reflects model differences. We do not claim it fully captures planning ability; rather, we view it as an observable outcome of planning in the travel domain.
>
> Specifically, satisfying complex travel constraints inherently requires reasoning depth, broad planning scope, and effective tool use. As the number of constraints increases, models can no longer solve the task by loosely stitching together itinerary fragments, but must perform stronger reasoning and global coordination. For example, when users specify preferences for different types of attractions and restaurants, the model must not only search for valid candidates, but also ensure the plan is coherent in time, space, and transportation—for instance, whether a restaurant that matches the user’s needs is actually available near a target attraction. This requires step-by-step search and computation, with constant switching between local decisions and global planning. If the resulting itinerary still fails to satisfy the constraints, the model must revise the plan as a whole, such as by replacing nearby restaurants or reordering attractions to improve transportation efficiency.
>
> We also analyze planning depth and reasoning horizon in Figure 3. The results show that performance is generally positively correlated with the number of tool calls and output tokens.
>
> ### Weakness 4
>
> We believe the highly challenging strict mode can be reserved for future, stronger models, since low success rates are common in difficult benchmarks. At the same time, loose mode already shows strong discriminative power across models, so the current two-mode design remains meaningful.
>
> ### Weakness 5
>
> We have already added stronger and more diverse baselines. Details can be found in our response to Reviewer ws2f, Weakness 1.
>
> ### Weakness 6 & Question 3
>
> We additionally use LongCat Flash Thinking 2601 and GPT-5.2 as user simulators and repeated the evaluation on DeepSeek models. Under loose / strict mode, DeepSeek achieves 37.0% / 41.0% and 9.5% / 9.8% pass rates, compared with 40.0% and 10.5% under the original setting. The differences are small, suggesting that with a sufficiently capable user simulator and well-designed prompts, simulation quality remains stable and does not materially affect fairness in cross-model comparison.
>
> ### Weakness 7
>
> We will make the corresponding corrections in the revision.
>
> ### Question 4
>
> For **local transport**, the model must ensure that all transportation transitions between relevant POI pairs are correct in both time and space. This is difficult to satisfy in one shot, but in the multi-turn setting, local revisions often also correct nearby transport arrangements, which likely explains why multi-turn performs better.
>
> For **hotel cost**, the multi-turn drop mainly comes from two issues: numerical hallucinations during hotel search, and later-added constraints that interfere with earlier hotel choices. In single-turn planning, hotels are often selected early, making the cost constraint easier to satisfy. In multi-turn interactions, models may fail to keep refining this constraint after a few unsuccessful attempts.

---

> > ### Author Rebuttal · Reviewer_cQHj · 2026-04-03
> >
> > Thank the authors for the detailed rebuttal. I keep my initial rating.

---

### Official Review · Reviewer_ws2f · 2026-03-13

**Soundness:** 3
**Presentation:** 3
**Significance:** 3
**Originality:** 3
**Overall Recommendation:** 5
**Confidence:** 3

**Summary:**

In this paper, the authors have created Trip-Bench a long-horizon benchmark that tests LLMs on multi-turn, real-world travel planning like feasibility transitions, ambiguous intent handling, and plan merging. Additionally, they propose GTPO method with three components: global instruction normalization, turn-wise reward differencing, and turn-level reward normalization to learn more effectively on these challenging tasks.
.

**Compliance With Llm Reviewing Policy:**

Affirmed.

**Final Justification:**

The rebuttal fully addressed my primary concerns. The expanded simulator validation (2.5x increase) and new baseline comparisons strengthen the paper considerably. The detailed failure mode breakdown adds value that should be incorporated into the final version.

**Key Questions For Authors:**

1. Strict subsets for all models show a big gap, for e.g. 44% vs 14% for GPT 5.2 in Hard LIT. So are there any specific types of constraint violations more common in each of these subsets? For e.g. in FIT subset, Is the primary failure in detecting infeasibility, in executing rollbacks correctly, or in satisfying constraints after rollback?
2. Could you provide more details on how "dialogue graph" graph is constructed and how the system prevents later changes from being visible to earlier steps?

3. Typos:
Line 32: "behavioral attributes))" to "behavioral attributes)"
Line 106: "Overall, our main contributions are as follows." appears twice
Line 436: "stabilizes training and improvr" to "stabilizes training and improves"
Line 435 col 2: "gaps in cross-turn consistency and meeting global constraints" to "gaps in cross-turn consistency and in meeting global constraints"

**Limitations:**

Explicit limitations of this benchmark are not discussed. For e.g. the work is limited to 40 cities within China and therefore has limited geographic diversity.

**Strengths And Weaknesses:**

## Strengths
1. The paper is well-motivated and very timely in terms of practical relevance. Also, the paper is generally well-written with a clear narrative.
2. I also like the fact the authors use paired generator-validator functions for each constraint type. Thus making the rewards verifiable unlike the many recent benchmarks that rely on LLM-based judges.
3. The four "hard" subsets (LIT, FIT, AIS, PMR) are creative and well-designed. These subsets capture realistic user behaviors like and are genuinely under-explored in most of the recent works.

## Weaknesses
1. The paper does not compare GTPO against other multi-turn RL methods it cites / even the most frequently used ones. I agree that GTPO seems to give strong gains against SFT, but how does it stand against other similar RL counterparts?
2. The manual evaluation of the user simulator reliability uses a relatively small sample: 20 trajectories for consistency checking and 10 trajectories for AIS-specific evaluation. Given that the entire evaluation pipeline depends on the user simulator, a larger validation sample would strengthen the paper.

---

> ### Author Rebuttal · Authors · 2026-03-29
>
> ### Weakness 1
>
> We would like to clarify that our work studies **joint reinforcement learning over both multi-turn tool interaction and multi-turn user interaction**, whereas existing methods usually cover only part of this setting.
>
> Most prior methods (e.g., Search-R1) only involve multi-turn tool interaction, without multi-turn user interaction, and typically use GRPO; this corresponds to **GRPO (ST)** in Table 3. Another line of work (e.g., MUA-RL) introduces multi-turn user interaction and rewards final task completion; this corresponds to **GRPO (MT)** in Table 3. Thus, the paper already compares against representative baselines from both categories.
>
> To further address this point, we additionally evaluated **DAPO** and **GSPO** under both ST and MT settings on **Qwen2.5-14B-Instruct**. The results are:
> GRPO (ST): **29 / 0 / 12 / 0**, DAPO (ST): **27 / 1 / 11 / 0**, GSPO (ST): **30 / 0 / 11 / 0**;
> GRPO (MT): **30 / 4 / 16 / 0**, DAPO (MT): **28 / 4 / 14 / 0**, GSPO (MT): **29 / 3 / 16 / 0**;
> while **GTPO (full)** reaches **35 / 13 / 18 / 0**.
>
> These results show that DAPO, GSPO, and GRPO perform similarly overall, while **GTPO** still maintains a clear advantage. This suggests that the gain does not come simply from changing the RL optimizer, but from modeling the **joint multi-turn user–tool interaction**.
>
>
>
> ### Weakness 2
>
> Under the multi-turn dialogue setting, **20 trajectories** correspond to **104 user-simulator generations**, and **10 AIS trajectories** correspond to **62 user-simulator generations**. Their effective coverage is therefore not comparable to 20 or 10 samples in a single-turn setting.
>
> To strengthen the evidence, we further expanded the human evaluation:
> the consistency check was increased from **20** to **50 trajectories**, covering **331 user-simulator generations**;
> the AIS-specific evaluation was increased from **10** to **25 trajectories**, covering **157 user-simulator generations**.
>
> With the expanded evaluation, the consistency check reaches **325/331 = 98.2%** reliability. In the AIS evaluation, the average score for accurately reflecting the intended ambiguity and faithfully simulating style is **4.74/5**. These results further support the reliability of the user simulator.
>
>
> ### Question 1
>
> Our additional analysis shows that the gap between **strict** and **loose** mainly comes from **near-miss cases rather than complete failures**. Many samples that pass loose but fail strict are already largely executable, but still miss one or two requirements on plan validity or user needs. In other words, loose accepts plans that are mostly correct but still have minor flaws, whereas strict does not.
>
> We manually analyzed hard cases for the two strongest models, **GPT-5.2** and **DeepSeek-V3.2**. Among cases that pass loose but fail strict, near-misses account for **76.1% (35/46)** for DeepSeek-V3.2 and **81.1% (38/47)** for GPT-5.2.
>
> Representative failure modes across the four hard subsets are:
>
> * **LIT**: **56%** are due to constraint drift over long conversations, i.e., forgetting previously stated constraints. About two-thirds involve ignored include/exclude constraints, while about one-third are small accumulated time or transportation errors.
> * **FIT**: **62%** are failures after rollback. The main issue is not failing to execute the rollback instruction itself, but failing to identify infeasibility or to maintain global consistency after constraints are revised.
> * **AIS**: **52%** involve misunderstanding the user request and failing to continue clarification; **34%** occur when the final plan still does not fully satisfy the clarified intent after user clarification or correction.
> * **PMR**: **48%** are caused by constraint confusion after plan revision or merging, such as old constraints leaking into the new plan or required constraints not being inherited correctly; about **8%** involve trip metadata not being properly updated after switching plans.
>
>
> ### Question 2
>
> For each evaluation plan, we first construct a set of reasonable transition links between constraints based on their dependency relations, and then form a directed graph. A later requirement becomes visible to the user simulator **only after its prerequisite requirements have been introduced in the dialogue**.
>
> For example, if we have:
>
> * (a → c)
> * (b → c)
> * (a → d → e)
>
> then **c** becomes visible only after both **a** and **b** have been introduced, and **e** becomes visible only after both **a** and **d** are satisfied.
>
> Therefore, later modifications are not exposed prematurely in earlier stages. At each turn, the user simulator only accesses dynamically updated information such as currently visible constraints, issue feedback, and dialogue history, but not future constraints that have not yet been unlocked.
>
>
> ### Question 3
>
> We will revise these spelling and phrasing issues throughout the paper for consistency.

---

> > ### Author Rebuttal · Reviewer_ws2f · 2026-04-01
> >
> > I thank the authors for their thorough rebuttal. My primary concerns have been adequately addressed.
> >
> > I am increasing my score from 4 (Weak Accept) to 5 (Accept). The original weaknesses have been resolved with new experimental evidence, and the failure mode analysis adds scientific value beyond what was in the submission.
> >
> > I would encourage the authors to incorporate the failure mode analysis (Question 1 response) and expanded simulator validation numbers into the final version, as these strengthen the paper considerably.

---

### Official Review · Reviewer_UhPh · 2026-03-13

**Soundness:** 1
**Presentation:** 2
**Significance:** 1
**Originality:** 1
**Overall Recommendation:** 2
**Confidence:** 4

**Summary:**

This paper introduces TRIP-Bench, a long-horizon benchmark for evaluating LLM-based interactive agents in realistic travel-planning scenarios, designed to capture challenges such as global constraint satisfaction, multi-tool coordination, and evolving user intent over extended dialogues. The authors further propose an online reinforcement learning method, GTPO, aiming to improve adherence to global rules and adaptation to dynamic preferences. The benchmark is well-motivated and large in scale, though its quality controls and positioning w.r.t. several recent travel-planning benchmarks could be clarified to better contextualize the contribution.

**Compliance With Llm Reviewing Policy:**

Affirmed.

**Key Questions For Authors:**

Please refer to the Weaknesses above.

**Limitations:**

No, authors should refer to Weaknesses #1-3 above.

**Strengths And Weaknesses:**

Strengths:

1. The paper is well-motivated, especially when it establishes a comparison between trajectories stemming from sharding (Laban et. al, 2025) and more challenging trajectories that involves rollbacks or shifts in user intent.

2. Dataset size is "around 120,000" examples (please report exact number) and benchmarking spans a broad set of backbone LLMs (thinking and non-thinking, open- and closed-source).

Weaknesses:

1. To properly contextualize their contributions, authors should cite "Flex-Planner: A Benchmark for Flexible Planning with Language Models" (June 2025), "RETAIL: Towards Real-world Travel Planning for Large Language Models" (Aug 2025), "DeepTravel: An End-to-End Agentic Reinforcement Learning Framework for Autonomous Travel Agents" (Sept 2025), and "TripTide: A Benchmark for Adaptive Travel Planning under Disruptions" (Oct 2025) due to substantial overlaps in introducing to the travel planning domain global constraints, ambiguous or shifting requirements, and plan adaptation needs over multiple turns.

2. Crucially, quality control is largely unclear. Specifically, from sub-section 3.3: "We sample full plans, evaluate them, and manually check whether flagged issues [by an LLM-as-a-judge] are repairable." How many samples were evaluated? How exactly? What is the proportion of samples containing unrepairable issues, such that this can be estimated to the entire dataset?

3. Certain design decisions are questionable, and it's difficult to determine to what extent can the proposed dataset serve as a realistic proxy for human behavior. Specifically, from sub-section 3.2: "We collect approximately 40 common requirement categories from real-world planning scenarios"---from where exactly? "For three-city cases, we retain only itineraries where (...) the three cities are roughly collinear, matching typical travel routes"---according to which data sources? Then, from sub-section 3.5, why are plan issues (detailed in Appendix B.4.2) such as "the schedule should be feasible with no overlaps" or "restaurant quantities and hotel room capacity must satisfy party-size requirements" categorized as "plan soundness" and therefore allowed to pass in the loose scenario, and not categorized as "basic feasibility" and therefore *not* allowed to pass in either scenario? Why wouldn't they be feasibility issues?

4. Presentation is confusing in a few parts, specifically: on Fig. 4 (left), why is "Cost" a "local constraint" and "Price" a "global constraint"? Also on page 8, Fig. 5 is part of an analysis but placed only 10 pages ahead, on page 18.

---

> ### Author Rebuttal · Authors · 2026-03-29
>
> ### Weakness 1
>
> We further clarify the main contributions of our work:
>
> 1. A **complete interactive environment** built on large-scale real-world data simulation, with 18 enhanced tools.
> 2. **Broad user coverage**: 40+ requirement types, 80+ expression patterns, and 9 representative multi-turn behaviors, including appending, modifying, redirecting, rollback, plan comparison/integration, local revision, error reporting, clarification/explanation, and exploratory inquiry. The hard subset further features longer dialogues (about 10 turns on average), ambiguous intents, style shifts, feasibility changes, and iterative revisions.
> 3. **Long-horizon tasks**: a single task may involve 150+ tool calls and up to 200k tokens of context.
> 4. **Complex rule and constraint reasoning**: models must plan under system rules and up to 12 constraints.
> 5. **GTPO**, an online RL method tailored for multi-turn tool use and user interaction, which significantly outperforms alternative training methods in our evaluation.
>
> Compared with prior work:
>
> * Flex-TravelPlanner mainly studies 2–3 turn interactions with single-constraint conflicts or priority adjustment, whereas our hard FIT subset targets long-horizon, multi-constraint, and rollback-intensive settings.
> * RETAIL focuses on implicit requirement clarification and gradual updates, similar to the user-bench in Table 1. Its context length is below 32k, and external information is provided by retrieval rather than the model’s own tool use, so it is not a complete agent benchmark. Methodologically, RETAIL uses prompt workflows and SFT on a 7B model, while our GTPO is RL-based and applied to 14B and 32B production-scale models in a much more complex setting.
> * TripTide studies plan revision after environmental changes, while we focus on sustained interactive planning under long-horizon, multi-turn, and complex constraints.
> * DeepTravel addresses single-turn travel planning, similar to the TravelPlanner in Table 1. In contrast, our work introduces both a more comprehensive benchmark and GTPO, specifically designed for long-horizon, multi-turn, multi-tool, and multi-user interaction.
>
> Overall, our work substantially extends prior studies in environment completeness, user behavior coverage, task horizon and complexity, and training paradigm.
>
> ### Weakness 2
>
> We apologize for the ambiguity. In the sentence *“We sample full plans, ...”* **sample** does **not** mean selecting a small subset of queries for inspection. Instead, for **every query in the evaluation set**, we repeatedly sample complete trajectories and evaluate them. Moreover, the evaluation is **not** based on LLM-as-a-judge; it relies entirely on automatic code execution. The process is:
>
> 1. For each seed query, we sample 4 trajectories from LongCat Flash Chat and 4 from DeepSeek v3.2, giving 8 trajectories in total.
> 2. We feed the code-based evaluation results back to the model and allow up to 3 rounds of iterative correction, similar to the SFT data construction process in Section 4.1.
> 3. If at least one trajectory passes all evaluations, we keep the query. Otherwise, we manually inspect the two trajectories with the fewest errors and judge whether the remaining errors are repairable.
>
> In the final evaluation set of 400 samples, about 40% come from the first case, where at least one trajectory passes after automatic correction, and about 60% require manual review to confirm repairability. This ensures that every query is **solvable in principle**, meaning that at least one valid solution can satisfy all evaluation criteria.
>
> ### Weakness 3
>
> First, for the **40 common requirement categories**, roughly half come from online data analysis and half from offline summaries based on around 15 real travelers, together providing reasonable coverage of common travel needs.
>
> Second, for **three-city cases**, we enumerate city triples from the 40 cities in our database, retain geographically plausible groups as candidate seeds, and manually verify route validity during query review.
>
> Finally, issues such as minor itinerary overlap or restaurant/hotel capacity mismatch are treated as rationality rather than feasibility, since they can usually be resolved through minor local adjustments, even by users themselves. In contrast, POI hallucination, clearly incorrect date arrangements, or broken transportation logic are basic feasibility failures because they make the plan non-executable and require major replanning.
>
> ### Weakness 4
>
> In Figure 4 (left), the distinction is defined by the **inner ring**: cost corresponds to hotels and is categorized as local, because hotel bookings are usually made only once or twice per trip; price corresponds to restaurants and is categorized as global, because dining expenses recur throughout the trip.
>
> For Figure 5, its current placement is mainly due to the page limit. In the revision, we will move it if space permits and add a brief explanation for clarity.

---

> > ### Author Rebuttal · Reviewer_UhPh · 2026-04-03
> >
> > I appreciate the reviewers' rebuttal. In particular, the clarifications pertaining to Weakenesses #1 and #2 are helpful and necessary for the paper. For example, sub-section 3.3 pasted below fully implies that quality control is restricted to the evaluation sets:
> >
> > 3.3. Quality Control
> > Although each component in our pipeline is solvable on its own, combining them can produce unrealistic cases
> > (e.g., preference–budget mismatches). We address this with prompt-based model scoring plus manual review. Because
> > travel plans have spatiotemporal dependencies—local feasibility doesn’t ensure global feasibility—we sample full plans, evaluate them, and manually check whether flagged issues are repairable. This two-stage validation keeps tasks practical and globally feasible.
> >
> > Respectfully, however, in my understanding Weakness #3 continues to be critical (e.g., from the rebuttal: "40 common requirement categories, roughly half come from online data analysis and half from offline summaries based on around 15 real travelers, together providing reasonable coverage of common travel needs.").
> >
> > Why and how are the design decisions underpinning this synthetic dataset representative of real users?
> >
> > I don't see sufficient evidence in how the benchmark was designed that performance on this dataset would be meaningfully informative of real-world performance---which, again, is critical for the utility of any benchmark. With other travel planning datasets being proposed and increasingly used, I don't see how this one rises above the others in terms of real-world utility.
> >
> > Due to the significance of Weakness #3, I choose to maintain my score.

---

> > > ### Author Response · Authors · 2026-04-03
> > >
> > > Regarding **benchmark construction**, we have already ensured that the **test set is solvable**, which we believe is the key requirement for a benchmark and is sufficient here. The training set is used only for SFT cold start. Specifically, rather than relying directly on raw training annotations, we conduct large-scale rollouts and retain only **fully correct** trajectories for SFT. In total, we generated about 120k trajectories and filtered them to around 3k fully correct, reasonable ones. We describe this process in Section 4.1 (Data Construction).  Therefore, we are unclear about your specific concern regarding “validation only on the evaluation set.”
> > >
> > > For the second point, we also **disagree that atomic needs collected from real online users and offline travelers would not be representative of real user needs**. Composing simulated users from such real atomic needs is a standard practice in travel planning literature, including TravelPlanner, Flex-Planner, and RETAIL. Compared with prior work, we not only substantially **expand the coverage of user needs (from around 10 to 40), but also provide a code-based evaluation function for each need type to enable precise evaluation**.
> > >
> > > More importantly, our benchmark goes beyond existing travel planning and user interaction datasets by explicitly modeling a class of capabilities that remains **largely missing from prior benchmarks, yet is essential for real-world agent deployment: diverse and challenging user behaviors**. The design of our hard subsets was also explicitly recognized by Reviewer ws2f:
> > >
> > > > *“The four ‘hard’ subsets (LIT, FIT, AIS, PMR) are creative and well-designed. These subsets capture realistic user behaviors and are genuinely under-explored in most of the recent works.”*
> > >
> > > This is also consistent with Reviewer cQHj’s assessment:
> > >
> > > > *“This work integrates complex constraint satisfaction with dynamic multi-turn user interactions. It specifically evaluates the capability of agents to satisfy complex constraints in long-horizon, multi-turn interactions—a capability essential for real-world agent deployment and sustained human-agent interaction, yet notably underrepresented in existing benchmarks.”*
> > >
> > > Taken together, we believe our dataset design and training method already provide strong evidence that this benchmark is scientifically valuable for understanding model capabilities in **realistic long-horizon interactive settings**. Specifically:
> > >
> > > 1. **TRIP-Bench targets real long-horizon bottlenecks, not just a harder synthetic task.**
> > >    It evaluates long conversations, underspecified requests, revisions, rollback, plan merge, and global consistency under multiple constraints.
> > >
> > > 2. **The core gap between single-turn and multi-turn agents is global consistency across turns.**
> > >    The main challenge is tracking and updating cross-turn state, not just executing isolated tool steps.
> > >
> > > 3. **Multi-turn interaction mostly turns near-misses into failures.**
> > >    Many models generate broadly executable plans but still miss a few critical requirements during interaction, making them locally usable yet globally unstable. The subsets expose different failure modes: LIT for constraint drift, FIT for rollback recovery or infeasibility recognition failures, AIS for intent understanding or clarified-need satisfaction failures, and PMR for confusion after plan revision or merge.
> > >
> > > 4. **Long-horizon performance depends on sustained reasoning and planning.**
> > >    Better results correlate with more tool-use turns, longer outputs, and explicit reasoning.
> > >
> > > 5. **These findings motivate GTPO as an effective online RL approach for long-horizon training.**
> > >    GTPO jointly models user interaction and tool interaction, better matches the training needs of long-horizon agents.
> > >
> > > Overall, our work is meaningful not only for the travel planning domain, but also beyond it, because it reflects broader capabilities required by real-world agents: sustained reasoning, interaction, and planning under long-horizon tasks, complex instructions, and diverse user behaviors. These are core agent capabilities and are directly important for improving real user experience. In addition, our training method **provides an effective online RL recipe for robust long-horizon training**. Moreover, regarding the distinction from other travel planning datasets, we have already explained in detail in our rebuttal to **Weakness 1** how our work differs from prior papers. In particular, we substantially extend prior work in terms of **environment completeness, user behavior coverage, task horizon and complexity, and training paradigm**. Therefore, we respectfully disagree with the claim that our work has limited real-world relevance or impact, or that it is not meaningfully different from prior work.

---

### Decision · Program_Chairs · 2026-04-30

**Decision:**

Accept (regular)

**Comment:**

This paper received overall positive but somewhat mixed feedback, with reviewers recognizing several key strengths:

* The paper introduces TRIP-Bench, a well-motivated and large-scale benchmark that captures realistic challenges in long-horizon, multi-turn travel planning, including global constraints, evolving user intent, and complex tool interactions.
* The benchmark design is comprehensive and thoughtfully constructed, with diverse task settings and difficulty levels that reveal meaningful limitations of current LLM agents.
* The proposed GTPO method is clearly formulated and demonstrates consistent improvements over baseline approaches, providing a useful step toward better handling multi-turn constraint satisfaction.

At the same time, reviewers noted several areas for improvement. The positioning of TRIP-Bench relative to existing travel-planning benchmarks could be more thorough, and additional clarity on dataset construction, quality control, and “real-world” grounding would strengthen confidence in its value. On the modeling side, broader comparisons to stronger RL baselines and more detailed analysis—such as failure modes, evaluation biases, and computational efficiency—would further support the claims.

During the rebuttal, the authors addressed several important concerns, including improving simulator validation, adding stronger baseline comparisons, and clarifying failure attribution, which increased confidence in the empirical results and evaluation pipeline. However, some issues remain only partially resolved, particularly regarding dataset positioning within prior work, detailed transparency of data construction, and the extent to which the benchmark reflects real-world user behavior.

Overall, the paper presents a valuable and timely contribution with a challenging benchmark and a promising method, though some aspects could be further strengthened. Therefore, I recommend weak acceptance.